# FL-GNN: A FUZZY-LOGIC GRAPH NEURAL NETWORK

## ABSTRACT

This paper presents a novel hybrid Fuzzy-Logic Graph Neural Network (FL-GNN) by combining Fuzzy Neural Network (FNN) with Graph Neural Network (GNN) to effectively capture and aggregate local information flows within graph structural data. FL-GNN by design has three novel features. First, we introduce a specific structure fuzzy rule to boost the graph inference capability of FL-GNN to be on par with the representative GNN models. Second, we enhance the interpretability of FL-GNN by adding the analytic exploration methods to its graph inference ability from two perspectives: Fuzzy Inference System and Message Passing Algorithm (MPA). Finally, we ameliorate the structure of FL-GNN based on MPA to address the inherent limitations of FL-GNN. This optimization can reduce the calculation complexity of FL-GNN and further improve its learning efficiency. Extensive experiments are conducted to validate the graph inference capability of FL-GNN and report the performance comparison against other widely used GNN models. The results demonstrate that FL-GNN can outperform existing representative graph neural networks for graph inference tasks.

## 1 INTRODUCTION

Graph is a powerful mathematical tool to model complex relationships between data items, such as establishing social networks, knowledge graphs, biological protein structures, etc. However, graph data often contain unknown or missing information that must be inferred. The process of inferring this information is known as graph inference, which encompasses a variety of tasks, including node classification, link prediction, graph generation, and so on. Graph inference has many applications in various domains, such as traffic-flow prediction, computer vision, and bioinformatics.

GNNs (Xu et al., 2019; Shi et al., 2021; Brody et al., 2022; Velickovic et al., 2018; Kipf & Welling, 2017; Hamilton et al., 2017) are a powerful tool in the deep learning domain to work with graph data and also the most popular tool for graph inference. GNNs learn the structure and feature information among the vertices and edges in the graph by iteratively updating their features based on their local or global neighborhood or even higher-order topology structure. This enables GNNs to capture the rich semantics and structural information hidden in graph data to apply to various graph inference tasks.

Fuzzy logic is a type of multi-valued logic that allows for degrees of truth instead of only crisp values. A fuzzy inference system (Czogala & Leski, 2000) is built using fuzzy logic consisting of a well-designed fuzzy rule base and a fuzzy inference engine, and it is often used to simulate human reasoning and decision-making processes. The fuzzy rule base comprises a set of IF-THEN fuzzy rules, such as "IF temperature is cold, THEN fan speed is slow". The quality of a fuzzy inference system depends on the design of the fuzzy rules and the choice of the membership function type and parameters, which are usually determined by experts. However, this task can be challenging and trivial for complex systems. To overcome this challenge, Buckley & Hayashi (1994) propose FNN, which draws lessons from Artificial Neural Network (ANN) to learn the parameters of a fuzzy system. Combining FNN with Deep Learning (DL) is a promising direction that has attracted extensive attention (Zheng et al., 2022; Das et al., 2021). Actually, FNN has been successfully applied in various fields such as medical image processing (Kaur & Singh, 2020), time series analysis (Luo et al., 2019), reinforcement learning (Fang et al., 2023), and multimodal sentiment analysis (Chaturvedi et al., 2019).

The current GNN has three limitations that can be addressed by exploiting FNN:

1) Representation capacity. The concept of fuzzification has been extensively researched to improve data representation capabilities. Real-world data often contains uncertainty and fuzziness beyond the scope of traditional crisp values (Hu et al., 2023; Krleza & Fertalj, 2017). To tackle this challenge, the tools for fuzzification have been developed to capture such fuzziness and uncertainty. However, relying solely on fuzzification as a data augmentation technique is insufficient to fully utilize the benefits of fuzziness. This is why traditional GNN frameworks are limited in their feature extraction, focusing solely on feature dimensions and disregarding the fuzzy dimension. To overcome this limitation, we have created the FL-GNN, which utilizes fuzzy rules to highlight fuzzy-level features and harness the advantages of fuzzy features for a more comprehensive data representation solution.

2) Interpretability. In traditional GNNs, the correlation between the network's parameter weights and inference process is implicit. In FL-GNN, the differences in topological structure between communities are explicitly presented in the firing strength distribution. By studying the firing strength distribution, we can know which rules are valid and which are redundant. Therefore, in this paper, interpretability not only provides a visual perspective for us to observe the local topology information and feature distribution differences of vertices but also provides reliable evidence for us to improve the model structure.

3) Degree of freedom. Several frameworks have been developed in graph inference, such as Message Passing Algorithms, which are compatible with most graph inference models. Fuzzy inference systems can offer a more flexible way to establish inference models. Fuzzy rules are not limited to graph-structured data and can be extended to other data structures and temporal data processing (Luo et al., 2019). In addition, in Hu et al. (2023), fuzzy rules are directly designed for specialized graph-level task inference. Therefore, we believe that the potential of FL-GNN will be fully realized in future works, where more effective fuzzy rules can be designed for graph inference tasks. In Appendix A, we have mentioned that the fuzzy rule of FL-GNN is limited to message aggregation in 1-order neighbors. Nevertheless, by designing more sophisticated fuzzy rules, they can be extended to complex topology structures such as "Hyperedge" and "Simplicial Complex", which will help the model break through the 1-dim WL-test.

Recently, there have been studies combining FNN with GNN. In Zhang et al. (2023), the authors propose using fuzzy features to carry out graph contrastive learning. The paper demonstrates the advantages of data fuzzification in representing graph information and supports this viewpoint with abundant experiments. However, the authors do not incorporate the core of fuzzy inference systems, i.e., the rule base and inference engine, into such graph inference tasks. In Hu et al. (2023), the authors propose a model to deal with graph-level inference tasks, including graph regression tasks and graph classification tasks. The model refers to the fuzzy logic system to generate appropriate fuzzy rules to adapt specific inference tasks. The authors utilize the graph cluster algorithm to find the prototype graphs, and each prototype graph's feature will be used to generate an IF-part of the fuzzy rule. Then, the prototype graph will be fed into a GNN-dominated network structure GCPU to generate the network parameters for the THEN-part of the fuzzy rule. This work focuses on graph-level inference tasks but ignores the local topology information, while the model utilizes graph kernel function and traditional GNN to extract the graph information instead of fuzzy logic.

One of the biggest challenges for integrating FNN and GNN is the discrepancy between fuzzy rules and graph inference patterns. fuzzy rules are usually designed to capture human intuition reasoning processes, while graph inference patterns are derived from high-level abstractions of graph structures. Hence, it is challenging to define effective fuzzy rules that work well with graph inference patterns. To address the challenge, we explore a new perspective, where the fuzzy rule can also be viewed as a flexible network structure. By defining a specific fuzzy rule, FNN can handle tasks with a unique data structure, such as the temporal relationship in time series data, the local relationship in image data, or even the topology relationship in graph data. With this in mind, a novel model FL-GNN is proposed, which bridges the gap between FNN and GNN by allowing both fuzzy rules and graph inference patterns to be represented in a unified way.

FL-GNN follows the fuzzy inference process with a human-like mindset and uses MPA (Gilmer et al., 2017) to explain it in the GNN field. From the perspective of the fuzzy inference system, FL-GNN utilizes the IF-part of the fuzzy rule to construct the fuzzy relation about the center vertex with its neighborhood vertices, and then the THEN-part outputs the defuzzification inference result as the high-order features for the center vertex. We can directly obtain the structure and semantic information for each vertex in the graph through the firing strength distribution. From the

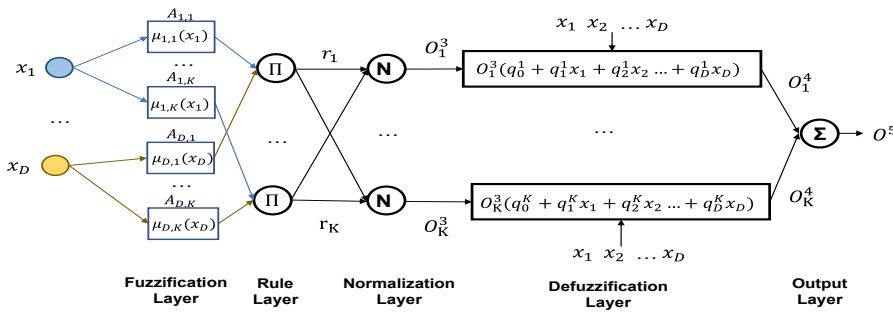

Figure 1: TS-FNN architecture

perspective of MPA, the IF-part is designed to capture and aggregate the neighborhood information by combining the t-norm operator and s-norm operator, and the THEN-part is designed to produce the representations by applying a linear transformation. The rule layer of FL-GNN can be seen as a set of aggregation functions to generate abundant aggregation results from different views, i.e., different permutation and combination methods of fuzzy subsets in the rule layer.

Our main contributions include: 1) This paper proposes a novel hybrid model named FL-GNN, which integrates the architecture of the FNN and the concept of MPA to handle various graph inference tasks. We also present two perspectives through which FL-GNN can be interpreted: MPA and fuzzy inference system. 2) An improved version of FL-GNN, FL-GNN-A, is proposed that significantly reduces model complexity while maintaining model performance. 3) Extensive experiments show the inference capability, performance, principle, and interpretability of FL-GNN and FL-GNN-A.

## 2 PRELIMINARIES:TAKAGI-SUGENO-FNN

Takagi-Sugeno-FNN (TS-FNN) (Rajurkar & Verma, 2017) is one of the most common FNN, whose architecture is shown in Figure 1, usually consists of 5 layers: fuzzification layer, rule layer, normalization layer, defuzzification layer, and output layer.

Given an input vector $\mathbf{x} = [x_1, .., x_D] \in \mathbb{R}^D$. The fuzzy subset corresponding to the $i$th input variable $x_i$ in the $k$th rule could be denoted as $A_{i,k}$. Let $\mu_{i,k}$ be the corresponding membership function (MF) of $A_{i,k}$. A Gaussian-type MF is defined as

$$\mu_{i,k}(x_i) = e^{\frac{-(x_i - c_{i,k})^2}{2\sigma_{i,k}^2}}, \tag{1}$$

where $c_{i,k}$ is the Gaussian function center, $\sigma_{i,k}$ is the Gaussian function width. They can be tuned according to the distribution of input data. When we use singleton fuzzification to fuzzify each input component $x_i$, the fuzzification result of $x_i$ is denoted by $\mathbf{o_i^1} = [\mu_{i,1}(x_i), ..., \mu_{i,K}(x_i)] \in \mathbb{R}^K$, and the output of the fuzzification layer is given by

$$\mathbf{O^1} = [\mathbf{o_1^1}, \mathbf{o_2^1}, ..., \mathbf{o_D^1}] \in \mathbb{R}^{D \times K}, \tag{2}$$

where $K$ is the number of rules. In the TS-FNN, we define the $k$th rule as follows:

$$IF\ x_1\ is\ A_{1,k}\ AND\ x_2\ is\ A_{2,k}\ ...\ AND\ x_D\ is\ A_{D,k}$$
$$THEN\ y^k(\mathbf{x'}) = q_0^k + q_1^k x_1 + ... + q_D^k x_D, \tag{3}$$

where, $\mathbf{q}^k = [q_0^k, q_1^k, ..., q_D^k]^\top$ $(1 \le k \le K)$ is the trainable parameters of defuzzification layer for the $k$th rule. "AND" is the t-norm operator in fuzzy logic, denoting the logic conjunction operation between fuzzy subsets. In addition, each t-norm in fuzzy logic has a dual operator named s-norm, which can be written as "OR", denoting disjunction operation between fuzzy subsets. In the following sections, we use $s(\cdot)$ and $t(\cdot)$ as the abbreviation for the s-norm and t-norm functions. As Equation (3) shows, the fuzzy rule is often built by IF-THEN structure. The IF-part of each rule generates a single value in the rule layer named firing strength, reflecting the matching degree of input

with the rule. Moreover, the THEN-part is responsible for defuzzification, which means converting the fuzzy value into a crisp value. TS-FNN calculates THEN-part on the defuzzification layer.

Here for Equation (3) if we use "product" as the t-norm operator, then the output of $k$th rule in the rule layer is $r_k = \prod_{i=1}^{D} \mu_{i,k}(x_i)$ $(1 \leq k \leq K)$. We call $r_k$ as the firing strength value. The output of the rule layer is the firing strength vector as

$$\mathbf{o^2} = [r_1, r_2, ..., r_K] \in \mathbb{R}^K. \tag{4}$$

The normalization layer is in charge of calculating the weight of the firing strength value of each rule within the whole fuzzy rule base, reflecting its importance of a reasoning process. Then, the $k$th rule is normalized as follows:

$$O_k^3 = \overline{r_k} = \frac{\prod_{i=1}^{D} \mu_{i,k}(x_i)}{\sum_{k=1}^{K} \prod_{i=1}^{D} \mu_{i,k}(x_i)}. \tag{5}$$

Thereafter, the normalized result of the rule layer is given by

$$\mathbf{o^3} = [O_1^3, O_2^3, ..., O_K^3] \in \mathbb{R}^K. \tag{6}$$

The defuzzification layer is designed to calculate the THEN-part of the fuzzy rule to output crisp value directly. The defuzzification result of the $k$th rule is given by

$$O_k^4 = O_k^3(q_0^k + q_1^k x_1 + ... + q_D^k x_D) = O_k^3(\mathbf{x}'\mathbf{q}^k), \tag{7}$$

where $\mathbf{x}'$ is the input vector in the defuzzification layer. In 1-order TS-FNN, $\mathbf{x}'$ is the ordinary input vector of the fuzzification layer concatenating extra element 1, $\mathbf{x}' = [1, \mathbf{x}] = [1, x_1, .., x_D] \in \mathbb{R}^{D+1}$. Besides, if $\mathbf{q}^k = [q_0^k]$, the TS-FNN will degenerate to 0-order while the input vector of the defuzzification layer becomes $\mathbf{x}' = [1] \in \mathbb{R}^1$. Furthermore, the above description just depicts a Multiple Input Multiple Output (MIMO) system; if we adjust the trainable parameter vector to $\mathbf{q}^k \in \mathbb{R}^{(D+1) \times out\_feature}$, the system will become to Multiple Input Single Output (MISO). The output vector of the defuzzification layer is denoted by

$$\mathbf{o^4} = [O_1^4, O_2^4, ...O_K^4] \in \mathbb{R}^K. \tag{8}$$

The output layer summarises the total output result of the defuzzification layer, i.e.,

$$O^5 = \sum_{k=1}^{K} O_k^4. \tag{9}$$

## 3 METHODOLOGY

We first introduce a new concept of fuzzy representation graph (FRG) to describe a graph with fuzzy and uncertain information. Then, we propose FL-GNN to conduct graph inference based on FRG.

### 3.1 FUZZY REPRESENTATION GRAPH

The information in the real-world graph may be incomplete and fuzzy. Fuzzifying the graph data can capture the fuzziness and prevent information loss. To this end, we present FRG (Fuzzy Representation Graph) to realize the node-level graph fuzzification.

FRG is denoted as $G = (V, E, F_v, F_e, A_v, A_e)$, where $V$ is the set of vertices, $E$ is the set of edges, $F_v$ is the vertex attribute set, $F_e$ is the edge attribute set, $A_v$ is the set of fuzzy subsets for the vertex attribute, and $A_e$ is the set of fuzzy subsets for the edge attribute. For $i$th attribute $F_{v,i} \in F_v$, we consider it as a universe of discourse, which has $k_{v,i}$ fuzzy subsets, corresponding to a membership function set $A_{v,i} = \{\mu_n, ..., \mu_{n+k_{v,i}-1}\} \subset A_v$. Similarly, for $j$th attribute $F_{e,j} \in F_e$, we also consider it as a universe of discourse, which includes $k_{e,j}$ fuzzy subsets, corresponding to a membership function set $A_{e,i} = \{\mu_m, ..., \mu_{m+k_{e,j}-1}\} \subset A_e$. Let $\phi : V \to 2^{F_v}, \psi : E \to 2^{F_e}$ denote the mapping functions for vertex attributes and edge attributes, where $2^{F_v}$ and $2^{F_e}$ are the power sets of $F_v$ and $F_e$, respectively, and then each vertex $v_i \in V$ and edge $e_i \in E$ can be represented by a set of vertex attributes $\phi(v_i) \subset F_v$ and a set of edge attributes $\psi(e_i) \subset F_e$. Meanwhile, we also define two fuzzy subset mapping functions, $\rho : 2^{F_v} \to 2^{A_v}, \sigma : 2^{F_e} \to 2^{A_e}$. Having the above functions, we could fuzzify any vertex and edge by mapping them into a set of fuzzy subsets in FRG, e.g., $\rho(\phi(v_i)) \subset A_v, \sigma(\psi(e_i)) \subset A_e$.

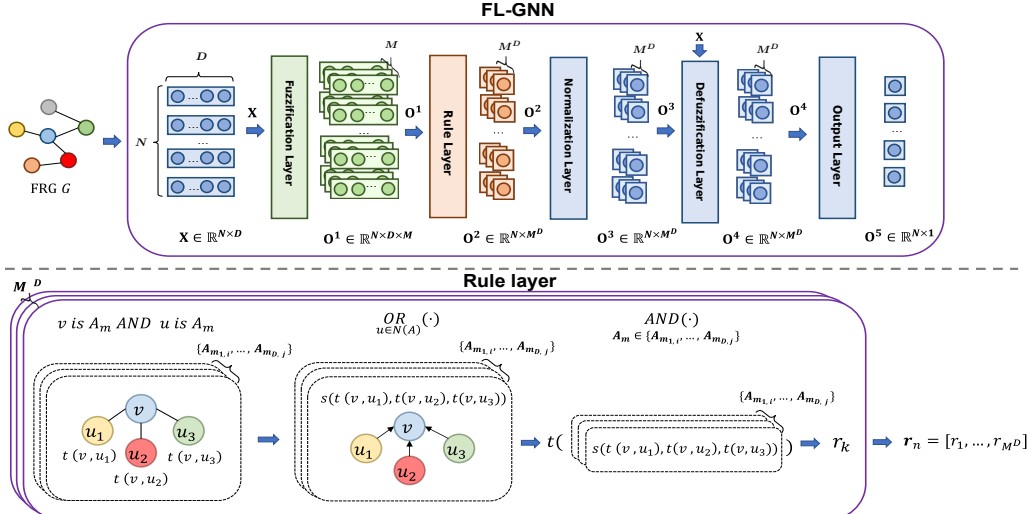

Figure 2: The architecture of FL-GNN

## 3.2 FL-GNN

The architecture of FL-GNN generally follows the TS-FNN. The difference between MIMO FL-GNN and MISO FL-GNN depends on the trainable parameters $\mathbf{q}^k$. Without loss of generality, we take MISO FL-GNN as an example to illustrate. The architecture of FL-GNN is shown in Figure 2. The upper box displays the main workflow of FL-GNN, while the lower box shows the working procedure of the rule layer for one vertex.

Given an FRG $G$ with $N$ vertices, each vertex's feature vector is denoted by $\mathbf{x}_n = [x_1, \ldots, x_D] \in \mathbb{R}^D$ ($1 \leq n \leq N$), and each attribute is assigned with $M$ fuzzy subsets, i.e., $|A_{v,d}| = M$ ($1 \leq d \leq D$). Let $A_{m_{i,j}}$ denote the $j$th fuzzy subset of $i$th vertex feature ($1 \leq i \leq D, 1 \leq j \leq M$), and the membership function of $A_{m_{i,j}}$ is denoted by $\mu_{m_{i,j}}$ (we use the abbreviations $A_m$ and $\mu_m$ in the remaining part). The input of FL-GNN is the collection of all vertices' feature vectors, $\mathbf{X} \in \mathbb{R}^{N \times D}$. These vectors are first expanded to the fuzzy dimension through the fuzzification layer and we obtain $\mathbf{O}^1 \in \mathbb{R}^{N \times D \times M}$. Then, in FL-GNN we write the $k$th ($1 \leq k \leq M^D$) rule of each vertex $v \in V$ as

$$
\begin{aligned}
IF \underset{A_m \in (A_{m_{1,a}}, \ldots, A_{m_{D,b}})}{AND} ( \underset{u \in N(v)}{OR} (v \text{ is } A_m \text{ } AND \text{ } u \text{ is } A_m)) \\
THEN \text{ } y^k(\mathbf{x}') = (q_0^k + q_1^k x_1 + \ldots + q_D^k x_D),
\end{aligned}
\tag{10}
$$

where the tuple $(A_{m_{1,a}}, \ldots, A_{m_{D,b}})$ is an element of $S = A_{v,1} \times A_{v,2} \times \cdots \times A_{v,D}$, and '$\times$' denotes the Cartesian product, and indexes $a, b$ represent a combination of index results generated by Cartesian product. Hence, set $S$ has a total of $M^D$ tuples, corresponding to $M^D$ rules in the rule layer.

The rule design is based on the concept of MPA to achieve aggregation and update. Specifically, the rule employs a compound logic expression in the IF-part to achieve a *Mixed Aggregation Function* (Beliakov et al., 2020), which can aggregate the local information in the fuzzy dimension. In the THEN-part, each vertex updates its self-state based on its firing strength value (aggregation message). The IF-part of Equation (10) corresponds to three steps inside the rule layer described in the lower box of Figure 2: 1) the neighboring vertices of the center vertex $v$ calculate their similarities to $v$ by applying the $t(\cdot)$ operator under the fuzzy subset $A_m$ Note that the expression $t(v, u_i)$ represents the t-norm operation on the fuzzy values of vertex $v$ and $u_i$ in the fuzzy subset $A_m$. 2) vertex $v$ uses the $s(\cdot)$ operator to aggregate the similarity messages from its neighboring vertices. 3) the aggregation results from different fuzzy subsets $(A_{m_{1,a}}, \ldots, A_{m_{D,b}})$ are further aggregated by $t(\cdot)$. Then, we obtain the ultimate aggregation outcome corresponding to the firing strength value $r_k$ ($1 \leq k \leq M^D$). The firing strength vector corresponding to the input vertex vector $\mathbf{x}_n$ is denoted as $\mathbf{r}_n = [r_1, \ldots, r_{M^D}]$ ($1 \leq n \leq N$) and the output of the rule layer is represented by

$\mathbf{O}^2 = [\mathbf{r}_1, \ldots, \mathbf{r}_N] \in \mathbb{R}^{N \times M^D}$. The working procedure of the rule layer is designed to implement an attention mechanism in the fuzzy dimension. Specifically, different combination results in $S$ will generate diverse attention scores to different fuzzy information. In Appendix C, we visualize the firing strength distribution values to verify that different vertices present different firing strength values (attention scores) for different rules. After normalization, the normalized firing strength values $\mathbf{O}^3 \in \mathbb{R}^{N \times M^D}$ and all vertices' feature vectors $\mathbf{X}$ are fed into the defuzzification layer to obtain the defuzzification result $\mathbf{O}^4 \in \mathbb{R}^{N \times M^D}$. Finally, the output layer performs the sum operation on $\mathbf{O}^4$ and outputs the final result $\mathbf{O}^5 \in \mathbb{R}^{N \times 1}$.

## 3.3 IMPROVEMENT TO FL-GNN

The aforementioned FL-GNN has two inherent limitations:

1) As the input feature dimension increases, the number of rules increases exponentially, directly leading to dimension explosion and information redundancy. Dimension explosion is a common problem faced by FNNs (Luo et al., 2019; Wang et al., 2022; Zhang et al., 2018; Wang, 2020; Yu et al., 2022). Wang et al. (2022) proposes to reduce the number of rules through constructing an OR-AND structure of fuzzy rule. Wang & Qiao (2022) propose a Restricted Boltzmann Machine (RBM) that can dynamically adjust the size of the hidden layer so as to reduce the dimension of input data. Besides, Yu et al. (2022) use the topology learning method to learn the data distribution in advance, enabling FNNs to self-organize more effective rule structures. Meanwhile, the number of rules with low firing strength values also increases with the number of rules, making it difficult to provide effective aggregation messages and causing information redundancy. For example, the cumulative firing strength distribution values on the ogbg-molhiv dataset indicates that almost 85% of the firing strength values are below 0.5, and 20% are below 0.2. The main reason is that as the number of rules grows, the granularity of fuzzy information that each rule focuses on becomes finer, making it difficult for the original data distribution to cover every fuzzy information that each rule targets.

2) The structural design of the defuzzification and output layers is still complex for graph inference tasks. The additional training parameters and computational steps introduced by the defuzzification and output layers are not proportional to the performance improvement they bring. Meanwhile, the firing strength vector is sufficient to reflect the local structure information, so we can directly use it as the representation information of the vertex in the graph. We visualize this phenomenon in Appendix B, where we use unsupervised learning methods to learn graph structure information only through the rule layer.

To address both limitations, we propose two solutions to improve our model as follows:

1) We first introduce a sliding window mechanism to achieve dimension reduction. Specifically, we use a sliding window in FL-GNN to split the high-dimensional input into multiple low-dimensional segments, and each individual segment is assigned an independent fuzzification layer and rule layer. This structural design allows multiple FL-GNNs to share the complexity of the monolithic FL-GNN, and the number of rules is reduced from $M^D$ to $BM^W$, where $W$ is the sliding window size ($W \ll D$), and $B$ is the number of segments determined by the sliding window stride size and $W$. Note that the calculation procedure of firing strength value for each input segment is absolutely independent, enabling us to design a parallel architecture that can process all segments simultaneously, thereby improving computation efficiency. Then, we propose a feature-refinement scheme to overcome the problem of information redundancy. The scheme chooses the MaxPooling1D function as the firing strength refiner to compress the firing strength vector by filtering out invalidated features to obtain a more expressive firing strength vector.

2) To improve the model efficiency for graph inference tasks, we make some changes to FL-GNN's architecture, replacing the defuzzification layer, normalization layer, and output layer with the refinement layer, concatenate layer, and fully connected layer. The refinement layer compresses the size of the firing strength vector to reduce information redundancy. The concatenate layer combines the vertex vector with its firing strength, allowing us to obtain more complete vertex feature information and local structure information. The fully connected layer offers the necessary trainable parameters and transforms the firing strength vector to the required form for downstream tasks.

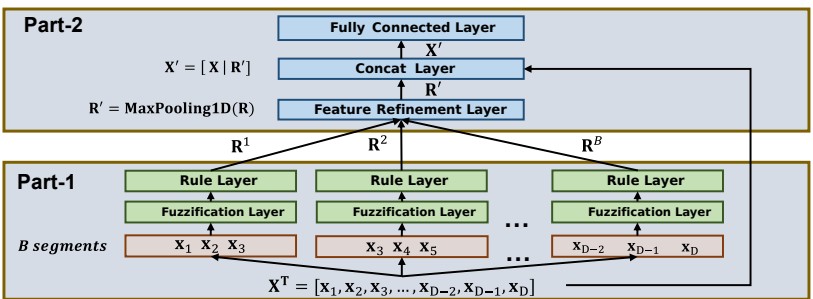

Figure 3: The architecture of FL-GNN-A

We incorporate two improvement solutions into FL-GNN and elicit FL-GNN-A. For FL-GNN-A, we use a two-part architecture to replace the original 5-layer architecture of FL-GNN to represent the aggregation operation and update operation in turn. The schematic diagram of FL-GNN-A is shown in Figure 3, where the sliding window size is 3, and the stride size is 2. To facilitate the description, we transpose the model input $\mathbf{X} \in \mathbb{R}^{N \times D}$ to $\mathbf{X}^{\mathsf{T}} = [\mathbf{x}_1, \mathbf{x}_2, ..., \mathbf{x}_D] \in \mathbb{R}^{D \times N}$, where the $\mathbf{x}_d \in \mathbb{R}^N$ $(1 \leq d \leq D)$ denotes the vector consisting of the $d$th feature of all vertices.

In part-1 of FL-GNN-A, we first utilize the sliding window to split the vertex feature along the feature dimension. These segments are then independently processed in the fuzzification layer and rule layer to generate the firing strength segment $\mathbf{R}^b = [\mathbf{r}_1^b, \ldots, \mathbf{r}_N^b] \in \mathbb{R}^{N \times M^W}$ $(1 \leq b \leq B)$, where $\mathbf{r}_n^b \in \mathbb{R}^{M^W}$ $(1 \leq n \leq N)$ denotes the firing strength vector for the $n$th vertex in the $b$th segment. The output of part-1 is denoted by $\mathbf{R} = [\mathbf{R}^1, \ldots, \mathbf{R}^B] \in \mathbb{R}^{N \times BM^W}$. After that, $\mathbf{R}$ is fed into the feature-refinement layer of part-2 to compress the size of $\mathbf{R}$ and obtain more informative firing strength $\mathbf{R}'$. In the concatenate layer, the original feature vectors and the firing strength values are combined to obtain $\mathbf{X}'$. Finally, we input the concatenate layer's output $\mathbf{X}'$ through a fully connected layer to generate the final outcome. Although FL-GNN-A does not follow the conventional structure of FNN, it still maintains the fundamental structure of the fuzzification layer and the rule layer. This implies that FL-GNN-A can effectively mitigate noise and extract fuzzy information from a graph. It is a trade-off between computational efficiency and the capacity for model inference.

# 4 EXPERIMENTS

## 4.1 DATASETS

To evaluate the performance of the model in various graph inference tasks, we conduct experiments on multiple datasets. Specifically, for node-level tasks, we choose three small node classification datasets, including Cora, Citeseer, and Pubmed (Yang et al., 2016), and two large-scale node classification datasets, Reddit (Hamilton et al., 2017) and ogbn-protines (Hu et al., 2020). Reddit uses F1-micro as its evaluation metric and ogbn-protines uses AUC-ROC as its evaluation metric. For the graph-level task, we choose three small-scale graph-level node classification/regression datasets, including ogbg-molhiv (classification), ogbg-molsol (regression), and ogbg-molfreesolv (regression), as well as the medium-scale dataset ogbg-molpcba (classification) (Hu et al., 2020). ogbg-molhiv uses ROC-AUC as an evaluation metric. The ogbg-molfreesolv and ogbg-molsol use RMSE, and ogbg-molpcba uses AP (average precision).

## 4.2 PERFORMANCE COMPARISON

**Experiment Settings.** For FL-GNN-A, we choose the "product" t-norm operator and replace its dual operator (s-norm operator) with the "mean" average operator by considering that the calculation of s-norm cannot be performed in parallel. We choose several widely-used GNNs as the baseline models, including Graph Isomorphism Network (GIN) (Xu et al., 2019), TransformerConv (Shi et al., 2021), GATv2 (Brody et al., 2022), GAT (Velickovic et al., 2018), Graph Convolutional Network (GCN) (Kipf & Welling, 2017), GraphSAGE (Hamilton et al., 2017). For the models with the multi-heads mechanism, such as TransformerConv and GAT, we set 6 heads for them. On the

Table 1: Performance on node-level dataset

| | Cora (F1-micro) | CiteSeer (F1-micro) | Pubmed (F1-micro) | Reddit (F1-micro) | ogbn-proteins (AUC-ROC) |
|---|---|---|---|---|---|
| FL-GNN-A | 0.8230±0.0039 | 0.7121±0.0065 | 0.7860±0.0010 | 0.9521±0.0015 | 0.7989 ±0.0043 |
| GraphSage-mean | 0.7930±0.0048 | 0.7206±0.0049 | 0.7689±0.0090 | 0.9501±0.0023 | 0.7962 ±0.0039 |
| GAT | **0.8506±0.0031** | **0.7368±0.0059** | 0.7852±0.0046 | 0.9429±0.0074 | 0.7996 ± 0.0018 |
| GCN | 0.8100±0.0028 | 0.7194±0.0034 | **0.7924±0.0420** | 0.9311±0.0131 | 0.7551±0.0079 |
| TransformerConv | 0.8002±0.0039 | 0.7160±0.0042 | 0.7803±0.0012 | **0.9550±0.0067** | **0.8003 ± 0.0025** |
| GATv2 | 0.8001±0.0036 | 0.7332±0.0024 | 0.7800±0.0023 | 0.9220±0.0445 | 0.7908 ± 0.0033 |
| GIN | 0.7950±0.0203 | 0.7001±0.0028 | 0.7512±0.0087 | 0.9218±0.0173 | 0.7569 ± 0.0054 |

Table 2: Performance on graph-level dataset

| | ogbg-molhiv (AUC-ROC) | ogbg-molesol (RMSE) | ogbg-molfreesolv (RMSE) | ogbg-molpcba (AP) |
|---|---|---|---|---|
| FL-GNN-A | **0.7863±0.0112** | 0.8113±0.0134 | 0.1821±0.0190 | **0.2468±0.0032** |
| GraphSage-mean | 0.7662±0.0419 | **0.7740±0.0205** | 0.1841±0.0280 | 0.2333±0.0045 |
| GAT | 0.7796±0.0223 | 0.8111±0.2638 | 0.1811±0.2007 | 0.2370±0.0632 |
| GCN | 0.7608±0.0247 | 0.7914±0.0074 | 0.1866±0.0309 | 0.2287±0.0113 |
| TransformerConv | 0.7816±0.0162 | 0.8325±0.0059 | 0.1823±0.2912 | 0.2433±0.0124 |
| GATv2 | 0.7504±0.0288 | 0.8277±0.0076 | 0.1794±0.0627 | 0.2455±0.0113 |
| GIN | 0.7803±0.0113 | 0.8101±0.0200 | **0.1787±0.0583** | 0.2423±0.0299 |

dataset Reddit, all models are trained in batches using the NeighborSampling scheme (Hamilton et al., 2017), and the sample sizes are set to $layer_1 = 35$ and $layer_2 = 20$ for both the 1st-order and 2nd-order neighbors, and on ogbn-proteins the sample sizes of the 1st-4th order neighbours are set to $layer_1 = 40$, $layer_2 = 35$, $layer_3 = 15$, $layer_4 = 5$. For the graph-level datasets, we select "sum" as the readout function.

**Experimental Results.** The results on node-level and graph-level graph inference tasks are displayed in Table 1 and 2, respectively. We observe that FL-GNN-A substantially possesses sufficient graph inference capability compared to popular GNN models. FL-GNN-A can handle both node-level and graph-level inference tasks regardless of their scales, indicating that the model structure of FL-GNN-A is suitable for common graph inference tasks. Furthermore, FL-GNN-A achieves great performance on graph-level datasets. For example, on both small-scale and medium-scale datasets, ogbg-molhiv and ogbg-molpcba, FL-GNN-A outperforms the other models. Meanwhile, FL-GNN-A still achieves good performance on the node-level tasks, where its performance exceeds most of the traditional models, especially on the Cora and Reddit datasets. We regard that the great performance of FL-GNN-A can be attributed to the fuzzification providing finer-grained features, which not only enhance the representation ability of the original features but also provide a certain extent of noise resistance, while the rule layer provides rich attention in fuzzy dimension, enabling FL-GNN-A to capture the fuzzy information more comprehensively.

## 4.3 ABLATION STUDY

Table 3: The performance results

| model | feature refine | hidden | ogbg-molhiv | hidden | Cora |
|---|---|---|---|---|---|
| FL-GNN | None | 5 | 0.7091±0.0171 | 5 | 0.3871±0.1181 |
| | | 7 | 0.7121±0.0165 | 7 | 0.5204±0.1048 |
| | | 9 | 0.7607±0.0165 | 9 | 0.6102±0.0581 |
| FL-GNN-A | None | 32 | 0.7496±0.2038 | 128 | 0.8138±0.0084 |
| | MaxPooling1D-70% | | 0.7616±0.0073 | | 0.8106±0.0097 |
| | MaxPooling1D-30% | | 0.7649±0.01584 | | 0.7918±0.0214 |
| | None | 64 | 0.7604±0.1769 | 256 | 0.8205±0.0056 |
| | MaxPooling1D-70% | | 0.7616±0.0273 | | 0.8183±0.0071 |
| | MaxPooling1D-30% | | 0.7768±0.0127 | | 0.8105±0.0062 |
| | None | 128 | 0.7840±0.0090 | 512 | 0.8213±0.0060 |
| | MaxPooling1D-70% | | 0.7723±0.0117 | | 0.8230±0.0039 |
| | MaxPooling1D-30% | | 0.7820±0.0101 | | 0.8192±0.0054 |
| FL-GNN-* | None | 32 | 0.6442±0.0106 | 128 | 0.3691±0.0349 |
| | | 64 | 0.6244±0.0201 | 256 | 0.4864±0.0182 |
| | | 128 | 0.6443±0.0198 | 512 | 0.4437±0.0261 |

Table 4: Model computing resource cost comparison

| hidden | FL-GNN-A† (layer=2) | | FL-GNN-A (layer=2) | |
|---|---|---|---|---|
| | Time consumption(ms) Inference/Training | Total trainable params | Time consumption(ms) Inference/Training | Total trainable params |
| 64 | 19.337 / 30.301 | 24,353,607 | 4.980 / 5.172 | 473,972 |
| 128 | 1907.848 / 4119.772 | 200,805,767 | 5.964 / 6.510 | 1,772,935 |
| 256 | NA | NA | 9.535 / 10.668 | 4,943,111 |
| 512 | NA | NA | 27.748 / 41.993 | 19,031,559 |
| hidden | FL-GNN-A† (layer=3) | | FL-GNN-A (layer=3) | |
| | Time consumption(ms) Inference/Training | Total trainable params | Time consumption(ms) Inference/Training | Total trainable params |
| 64 | 135.894 / 245.978 | 36,485,319 | 7.877 / 9.548 | 497,991 |
| 128 | NA | NA | 13.130 / 16.730 | 1,867,271 |
| 256 | NA | NA | 23.497 / 30.585 | 7,229,703 |
| 512 | NA | NA | 47.609 / 235.219 | 28,177,415 |

We conduct two groups of experiments to verify whether FL-GNN and FL-GNN-A positively impact graph inference and whether FL-GNN-A can improve the inference speed and reduce the trainable parameters of FL-GNN.

It is crucial to note that the **NA** in the experiment result indicates that the memory requirement exceeds the limitations of our hardware.

In the first group of experiments, we compare performance among FL-GNN, FL-GNN-A, and FL-GNN-* on the ogbg-molhiv and Cora datasets, where FL-GNN-* is a variant of FL-GNN-A that removes the fuzzy inference module and retains only the necessary fully connected layer. We use the feature-refinement module to extract 70% and 30% of the firing strength values, i.e., 70% and 30% of the length of the original firing strength vector will be retained. For the FL-GNN, when we set the number of membership functions for each feature greater than or equal to 3 and the number of hidden dimensions exceeds 10 the dimension explosion will occur. Since, in this experiment, we set the number of hidden units for FL-GNN to be much smaller than FL-GNN-A and FL-GNN-*. As Table 3 shows, compared to FL-GNN-*, FL-GNN achieves a significant performance improvement on the Cora and ogbg-molhiv datasets. This improvement is attributed to the fuzzy inference structure, which provides FL-GNN with powerful graph inference ability. Compared to FL-GNN, FL-GNN-A achieves further improvement. We regard that FL-GNN-A strikes a significant balance between model performance and computational overhead. However, for FL-GNN, the improvement in model performance is disproportionate to the increase in computational overhead of the model. In addition, when observing the effect on the MaxPooling1D function, we find that the MaxPooling1D function only slightly affects the final model performance. However, the MaxPooling1D function improves the speed of inference and training $1.2X \sim 1.3X$, while significantly reducing the trainable parameters of the model.

In the second group of experiments. We investigate the effect of using the part-2 architecture of FL-GNN-A on time consumption and the number of trainable parameters. We introduce an ablation model FL-GNN-A†, which retains the part-1 architecture but replaces the part-2 architecture with the normalization layer, defuzzification layer, and output layer same as FL-GNN. Also, we introduce a naive FL-GNN-A as a comparison, the feature refinement ratio of FL-GNN-A is 70%. For FL-GNN-A and FL-GNN-A†, we set the sliding window size to 5, the sliding step size to 5, and the number of membership functions for each feature to 3. Table 4 presents the comparison results of time consumption and variations in trainable parameters for FL-GNN-A† and FL-GNN-A on the Cora dataset under different hidden dimensions and stacked layers. From the table, it can be observed that FL-GNN-A† showcases an exponential increase in time consumption and trainable parameters as the hidden dimensionality increases. In comparison, FL-GNN-A shows a linear increase. The reason is that the defuzzification layer and output layer incur excessive computation overheads. However, FL-GNN-A utilizes the fully connected layer to replace them.

## 5 CONCLUSION

Integrating FNN with GNN offers new insights into graph inference using fuzzy logic. The fuzzy inference system provides an inference paradigm with high degrees of freedom and enhances the interpretability of reasoning processes by using formalized inference rules. Fuzzy set theory enables numerical representation of various data sources, improving the data fusion process. Currently, there are still some limitations in FL-GNN that need to be overcome in the future. 1) FL-GNN

uses Type-1 fuzzy sets to fuzzify the feature data of vertices, but the ability of Type-1 fuzzy sets to represent fuzziness is limited. Instead, more powerful fuzzy sets such as Type-2 fuzzy (Castillo & Melin, 2008) sets and intuitionistic fuzzy (Luo et al., 2019; Eyoh et al., 2018) sets can provide stronger fuzziness. By using these fuzzy sets, we can obtain better representation capabilities. 2) Computational complexity is still an issue in FL-GNN due to the need to handle a large number of fuzzy rules. Although we proposed FL-GNN-A in this paper to greatly alleviate this problem and enable FL-GNN-A to work effectively on large datasets, how to reduce computational complexity further remains a highly challenging issue. One direction we plan to pursue is to use the interpretability property of FL-GNN to guide us in designing evolutionary algorithms to compress network structures and obtain more effective models.

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

APPENDIX

# A  WHY FL-GNN IS USEFUL?

In this section, we explain the principle of FL-GNN.

Fuzzy neural networks are particularly attractive because of their interpretability. This means that the IF-THEN rules are made up of understandable linguistic variables, which makes it easier for humans to understand. FL-GNN can also be explained from the perspective of the fuzzy inference system. To illustrate this, let's consider a scenario where we have a social network graph of a campus, where each vertex has two attributes, namely "$age$" and "$income$". Our goal is to determine the career of each vertex based on these two attributes. Using common sense, we can come up with the following two appropriate rules:

(1): IF **most** of **the neighborhoods** surrounding $v$ are **young** in age **AND** their income is **low**, **THEN** $v$ is a student.

(2): IF **most** of **the neighborhoods** surrounding $v$ are **old OR middle** in age **AND** their income is **high**, **THEN** $v$ is a teacher.

The above two rules can be interpreted from the view of FL-GNN's fuzzy rule, where "age" and "income" are the universe of discourse, "**young**", "**middle**", and "**old**" are the linguistic label of fuzzy subsets of "age", "**low**" and "**high**" are the linguistic labels of fuzzy subsets of "income". The "**most**" indicates the high matching degree of the inner part of the fuzzy rule $OR_{u \in N(v)}(\cdot)$, which means the similarity between the center vertex $v$ and its neighborhoods. Here, we merely use the 1-hop neighborhood of the center vertex $v$ to define the concept "**neighborhood**", namely, the pairwise relationship between two vertices. To further ameliorate the performance of FL-GNN to breakthrough 1-WL test (Xu et al., 2019), we can also introduce the $higher\ order\ relationship$ concepts such as $Hyperedge$ or $Simplicial\ Complex$ to better interpret "**neighborhood**". Moreover, the innermost operator ($v\ is\ A_m\ AND\ u\ is\ A_m$) corresponds to the similarity of the vertex pair, which is the concept of "$Strength\ of\ Connectedness$" of "$Complete\ fuzzy\ graph$"(Mordeson et al., 2023), indicating the maximum value of the path strength between two vertices. In other words, based on the "similarity of vertex pair", we can filter out noisy vertices adjacent to the central vertex but with low correlation. Subsequently, "**AND**" integrates different factors included in reasoning to construct a complete rule, which corresponds to the outer part of the fuzzy rule $AND_{A_m \in (A_{m_{1,a}},...,A_{m_{D,b}})}(\cdot)$. Therefore, the inference result "**student**" or "**teacher**" can support more high-level inference, which interprets why FL-GNN can achieve better performance through a stacked structure. Additional experiments have been included in Appendix C to better showcase the interpretability of FL-GNN.

On the other hand, let us shift our focus to the Message Passing Neural Network (MPNN) (Gilmer et al., 2017), which is the generalized paradigm for most GNNs. MPA is a crucial phase of MPNN. The major steps of MPA can be divided into two steps: the message aggregation step and the message update step, where the step of message aggregation for each vertex combines the neighbors' message and leverages some aggregation mechanisms such as $sum$, $concatenate$, $attention$, etc., to aggregate them. The message update step for each vertex updates the self-state using some update function.

We consider the calculation process of the IF-part in each fuzzy rule as the *Mixed Aggregation Function* $f : [0,1]^n \rightarrow [0,1]$ to take real arguments from the closed interval $\mathbb{I} = [0,1]$ and produces a real value in that same interval. The components of $f$ that compose the t-norm and s-norm operators are the conjunctive aggregation function and disjunctive aggregation function, respectively (Beliakov et al., 2020). No matter whether t-norm or s-norm, they both possess $symmetric$, namely, $permuation\ equivariant$, which property satisfies the basic inductive bias of GNN. Moreover, we have also attempted to leverage the average aggregation function to supersede t-norm and s-norm. Because we consider when there are too many aggregated elements, the aggregation results of t-norm and s-norm tend to be the extreme value of 0 and 1, yet the averaging aggregation still leads to a technically valid aggregation result.

To make a summary, in IF-part FL-GNN generates $M^D$ different aggregation functions (fuzzy rules), and the THEN-part in each fuzzy rule plays the role of update function to combine the aggregation

message (firing strength value) into center vertex to generate new vertex state. In terms of the conclusion of Corso et al. (2020)): "*In order to discriminate between multisets of size n whose underlying set is* $\mathbb{R}$*, at least n aggregators are needed.*". FL-GNN obtains rich aggregation functions by combining different fuzzy subsets. Therefore, we believe that the rule layer provides enough inference capability for FL-GNN.

## B  FL-GNN AND MPA.

In order to concretize the relation between FL-GNN and MPA, We use an unsupervised learning approach to show how our fuzzy inference module captures the structural information of a graph without any label information, which also had been conducted in Kipf & Welling (2017). In this experiment, we select Zachary's Karate Club Network, a simple graph that is undirected and unweighted and only has the graph topology information. Then each vertex is labeled by one of four classes by the method of modularity-based clustering Brandes et al. (2008). In the beginning, the vertex vector is initiated by identity matrix $\mathbf{X} = I_N$, in which $N$ is the number of vertices. The identity matrix ensures that all graph structure information is learned from the model rather than obtained from vertex feature vectors. We select FL-GNN-A for feature extraction and set the number of membership functions in each feature to 3, the hidden dimensionality to 128, and the stacked layer to 3.

The initial distribution of each vertex is shown in Figure 4a, where the color denotes the community to which the vertex belongs, we can see that there is no distinct community distribution, suggesting that the identity matrix cannot provide any structural information. Therefore, the vertices of various classes cannot establish clear community boundaries. However, after undergoing feature extraction by FL-GNN-A, as shown in Figure 4b, the community boundaries become apparent. This reveals that the rule layer of FL-GNN is essentially extracting structural information from the graph.

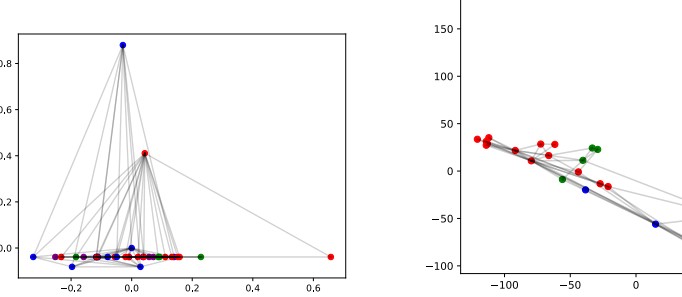

(a) Karate Club Network before embedding    (b) Karate Club Network after embedding

Figure 4: Graph embedding experiment on Karate Club Network graph

## C  INTERPRETABILITY OF FL-GNN.

In order to demonstrate FL-GNN's interpretability, we use a campus scenario mentioned in Appednix A. The campus social network consists of three types of roles: students, teachers, and logistics, and each vertex has two features: age and income. To make the social network graph reflective of reality, we use the Stochastic Block Model (SBM) algorithm Lee & Wilkinson (2019) to generate a stochastic graph that characterizes real-world graph properties such as power-law degree distribution, small diameter, and overlapping communities. The configuration of the stochastic campus network is listed in Table 5. We then set up FL-GNN with three fuzzy subsets for each feature. For the fuzzy subsets of age, the linguistic labels are *young*, *middle-age*, and *old*. For the fuzzy subsets of income, the linguistic labels are *low*, *medium*, and *high*. This configuration will generate $3^2 = 9$ rules in the rule layer. Next, we input the graph into FL-GNN (single layer, without training) and we can directly observe the firing strength distribution in the rule layer. We consider the differ-

ent firing strength distributions in the rule layer of each vertex can represent the unique identity in the social network, similarly, the vertices with identical identities exhibit analogous firing strength distributions in the rule layer.

In Figure 5a, the stochastic campus social network graph is generated using SBM, which includes students (blue), teachers (green), and logistics (red). In our code, the permutation of rules conforms to M-digit (M is the number of fuzzy subsets) encoding, specifically, each digit of the rule number points to a feature, and the value of this digit points to the fuzzy subset of the feature. In this case, we set three fuzzy subsets for each feature, thus the ternary encoding of the 7th rule is 21. The linguistic label corresponding to the 7th rule is "**income** is **high** AND **age** is **middle**". The ternary encoding of the 0th rule is 00. The linguistic label of the 0th rule is "**income** is **low** AND **age** is **young**".

The average firing strength of each career community with 9 rules is shown in Figure 5b. By observing which rules in each community have higher firing strength values, we can find the student community is adapted to Rule:0, the teacher community is adapted to Rule:7 and Rule:8, and the logistic community is adapted to Rule:4 and Rule:5. According to the linguistic label for each rule we could summarize the inference rule as follows

Table 5: Configuration of stochastic social network

| | | Graph Setting | | Vertex Feature | |
|---|---|---|---|---|---|
| | number | connection probability | | age | income |
| Student | 15000 | [Student:0.002, Teacher: 0.001, Logistic: 0.001] | | $X \sim U(15,25)$ | $X \sim U(0,2000)$ |
| Teacher | 300 | [Student:0.001, Teacher: 0.25, Logistic: 0.02] | | $X \sim U(35,70)$ | $X \sim U(9000,13000)$ |
| Logistic | 100 | [Student:0.001, Teacher: 0.02, Logistic: 0.25] | | $X \sim U(35,70)$ | $X \sim U(5000,6000)$ |

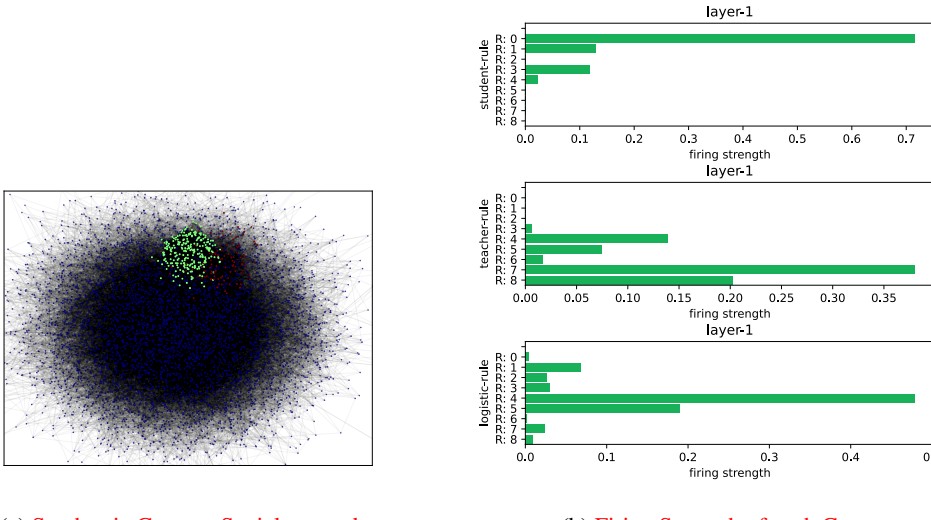

(a) Stochastic Campus Social network

(b) Firing Strength of each Career

Figure 5: The Stochastic Campus Social Network graph (left) and firing strength distribution for each career (right)

(1): IF **most of** $v's$ **neighborhoods'** age is **young**   AND income is **low**, **THEN** $v$ is student.

(2): IF **most of** $v's$ **neighborhoods'** age is **middle OR old**   AND income is **high**, **THEN** $v$ is teacher.

(3): IF **most of** $v's$ **neighborhoods'** age is **middle OR old**   AND income is **middle**, **THEN** $v$ is logistic.

In the above experiment, we only present narrow interpretability, where we assume a simple social network scenario where the raw features of each vertex are comprehensible so that we can confer the feature with meaningful linguistic labels to make the fuzzy rule intuitionally understood.

In FL-GNN, interpretability is not limited to linguistic labels. The interpretability of FL-GNN is based on hidden features. That is, the firing strength distribution of the fuzzy rule can provide interpretations for the inference process of FL-GNN. The firing strength distribution of communities directly reflects their topological structure information and feature distribution. Here topological information is specified as the degree of community overlap, i.e., the probability of having edges within a community and between multiple communities. Feature distribution specifically refers to the differences in node feature value between different communities. As we can see in the above experiment, communities of different professions exhibit intuitive differences in firing strength distribution.

In FL-GNN, the differences in topology structure between communities are explicitly presented in the firing strength distribution, indicating that there is a direct relationship between which fuzzy rules have a positive effect on capturing topology differences and which ones do not work. Therefore, our improvement in fuzzy rules can be intuitively reflected in the differences in model performance, just as we performed MaxPooling on the output of fuzzy rule layers in FL-GNN-A. Filtering out redundant rules not only ensures model accuracy but also improves the inference speed of the model.

In Section 3.2, we introduce a Multiple Input Single Output (MISO) FL-GNN. Here, we use Multiple Input Multiple Output (MIMO) FL-GNN for the generic case to introduce stacked FL-GNNs. The output of MIMO FL-GNN is defined as $\mathbf{O}^5 \in \mathbb{R}^{N \times hidden\_nums}$, where $N$ denotes the number of nodes. The difference between MISO FL-GNN and MIMO FL-GNN is that the value space of trainable parameters $\mathbf{q}^k$ of $k$th fuzzy rule changes from $\mathbb{R}^K$ to $\mathbb{R}^{K \times hidden\_nums}$, where $hidden\_nums$ is the hidden dimension size.

We formally define the FL-GNN with a stacked structure as follows: First, similar to the definition of FL-GNN in Section 3.2, a single-layer MIMO FL-GNN is defined as $\mathbf{O}^5 = f_\theta(\mathbf{X}, \mathbf{A})$, where $\mathbf{X}$ denotes the input node data, $\mathbf{A}$ denotes the adjacency matrix, $\theta$ denotes the trainable parameters in FL-GNN, and $\mathbf{O}^5$ denotes the output of the single layer of FL-GNN. In the following description, we replace $\mathbf{O}^5$ with $\mathbf{H}$ to make the formula more concise. Then, for the FL-GNN with a stacked structure, we define the output of the $l$th layer as $\mathbf{H}^l = f_{\theta^l}(\mathbf{H}^{l-1}, \mathbf{A}) + \mathbf{H}^{l-1}$, where we add the residual connection $\mathbf{H}^{l-1}$. Meanwhile, some research papers have reported that the FNN structure has the ability to fit nonlinear functions (Hu et al., 2023; Rodríguez-Fdez et al., 2016; Jang, 1993). To enhance the sparsity of the parameters and mitigate overfitting, we incorporate an activation function $\sigma(\cdot)$ of the ReLU family in each layer of FL-GNN. In addition, we also add a BatchNormalization Layer BN$(\cdot)$ for the stability of the model. Finally, the output of the $l$th layer is defined as $\mathbf{H}^l = \text{BN}(\sigma(f_{\theta^l}(\mathbf{H}^{l-1}, \mathbf{A}) + \mathbf{H}^{l-1}))$.

In Figure 6 we visualized the firing strength distribution of each layer in the FL-GNN with the stacked structure. By observing the changes in trends of firing strength distribution in different communities, we have found that as the interaction order increases, the firing strength distribution of a community gradually changes from reflecting the topology information of a single community to that of multiple overlapping communities. Ultimately, the firing strength distributions of all communities will tend to be consistent, representing the topology information of the entire graph. We can also notice that the two communities of teachers and logistics converge faster, while the student community converges relatively slower, due to the higher overlap (higher probability of edge connection) of the two communities of teachers and logistics, as a result, information is exchanged more quickly between the two communities. The student community has a lower overlap (lower probability of edge connection) with the teacher community and logistics community, and the student community has a much larger community radius. Therefore, nodes at the edge of the community need higher-order message passing to access information from the rest of the community.

For a more general scenario, we try to verify that fuzzy rules with low firing strength values are redundant for the FL-GNN inference process, which will further prove that the firing strength distribution is explicitly related to the graph inference process of FL-GNN. Therefore, we conducted fidelity test experiments to verify whether the fuzzy rules with low firing strength values are redundant for the inference of FL-GNN-A. We also introduced the GNNExplainer(Ying et al.,

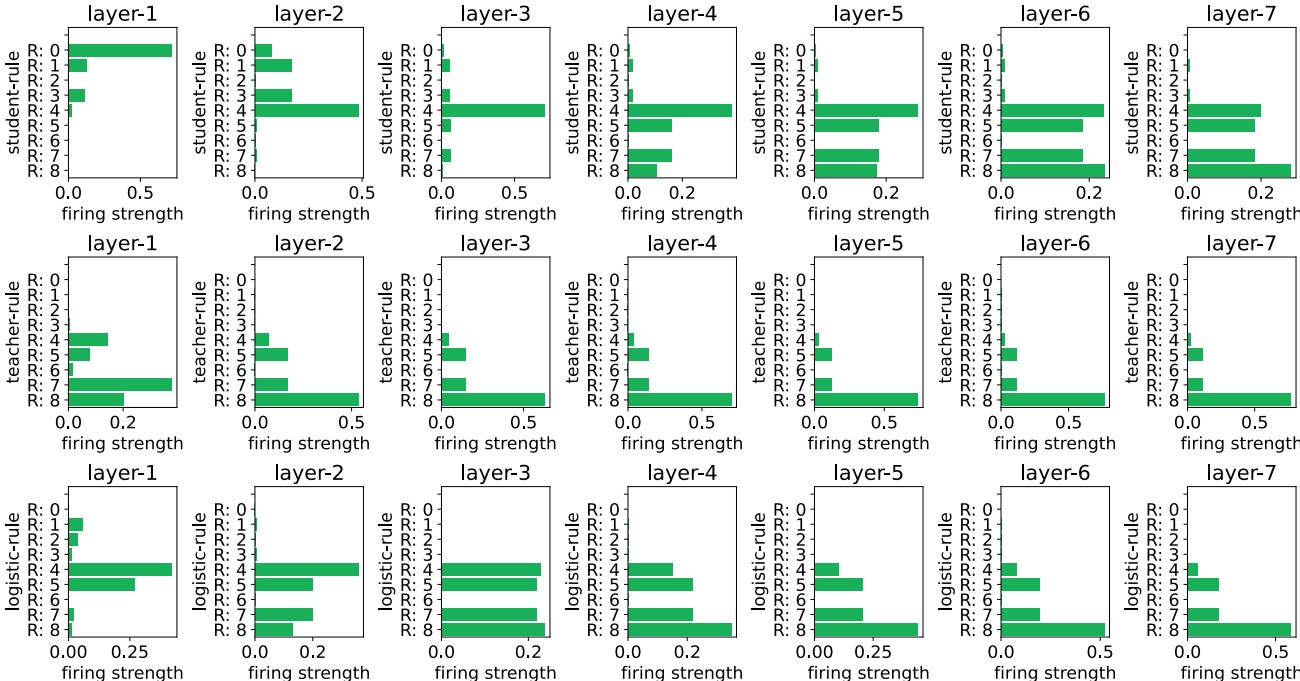

Figure 6: Firing strength distribution under indirect interaction of vertices. The three rows from top to bottom represent the firing strength distribution of students, teachers, and logistics. From left to right, it represents the firing strength distribution from 1-order to 7-order interaction.

2019) as a performance control. GnnExplainer is a model-free method that can provide a reliable explanation for GNN that conforms to the message-passing paradigm. Specifically, GnnExplainer will take a trained GNN model and its prediction results as input, and output the subgraphs and feature sets that truly affect the prediction results.

In the experiment section, we set up two sets of experiments. In the first group, we pre-trained the FL-GNN-A on the Pumbed dataset. Then we mark the indexes of the fuzzy rules that have low values in the inference phase of the pre-trained model. Next, we re-trained the model while masking out the corresponding fuzzy rules for the bottom 70%, 50%, and 30% of the ranked firing strength values in each layer. In the second experimental group, we feed the FL-GNN-A model pre-trained on the Pubmed dataset into the GNNExplainer model to extract the subgraphs as well as the feature sets that are more useful for the inference task. Next, the information extracted by GNNExplainer is used for inference on the test dataset to calculate the fidelity of GNNExplainer. For GNNExplainer the regularization hyperparameters for subgraph size is 0.005; for laplacian is 0.5; for feature explanation is 0.1. The experimental results are presented in Table 6.

From Table 6, we observe that when we mask out the fuzzy rules with firing values in the bottom 30%, the model's prediction performance improves, while when we mask out 50% of the fuzzy rules the model's performance only fluctuates slightly, which we believe is because the fuzzy rules with low firing strength values become noise in the prediction. When we mask out 70% of the fuzzy rules, the model performance decreases due to the loss of too much useful information. This shows that there is a clear correlation between the firing strength distribution and the graphical inference process of FL-GNN-A, and thus our masking of fuzzy rules with low firing strength values does not affect the inference process. At the same time, we can see that the information extracted by GNNExplainer improves the model significantly. We believe that the reason for such a difference is as follows: 1) The training of GNNExplainer is dependent on the prediction results, so the label information

in the test dataset is utilized for the training of the interpreter. While FL-GNN-A performs rule masking without utilizing the label information. 2) In the first group of the experiment, we only mask redundant rules without extracting more efficient rules among the high firing strength value ones. Nevertheless, by locating the redundant parts of the network, we can further simplify the network structure of FL-GNN-A, thereby further improving the inference efficiency of the model.

Table 6: The Fidelity Experiment on the Pubmed dataset.

| Pubmed | Pubmed-70% | Pubmed-50% | Pubmed-30% | GNNExplainer |
|---|---|---|---|---|
| $0.7704 \pm 0.0012$ | $0.7536 \pm 0.0090$ | $0.7733 \pm 0.0013$ | $0.7803 \pm 0.0019$ | $0.9397 \pm 0.0004$ |

