# OpenReview forum: "FL-GNN: A Fuzzy-logic Graph Neural Network"
_ICLR.cc/2024/Conference — Submitted to ICLR 2024_

### Official Review · Reviewer_CtCE · 2023-10-31

**Soundness:** 3 good
**Presentation:** 2 fair
**Contribution:** 2 fair
**Rating:** 5
**Confidence:** 3

**Summary:**

The paper introduces the FL-GNN, a fusion of Fuzzy Neural Network (FNN) and Graph Neural Network (GNN), aiming to harness the benefits of fuzzy logic in graph inference. The proposed model addresses key challenges in GNN, boosting its inference capabilities, enhancing interpretability, and improving efficiency by reducing computational complexity.

**Strengths:**

* Novel Approach: The amalgamation of FNN and GNN is an innovative step, seeking to combine the strengths of both paradigms for improved graph inference.

* Enhanced Interpretability: One significant contribution is the enhancement in model interpretability, a challenge with many machine learning models, especially neural networks.

**Weaknesses:**

* While the paper claims that FL-GNN outperforms other GNN models, it could benefit from a more rigorous benchmarking, including challenges faced, methodologies adopted, and potential pitfalls.

* It would benefit the paper to clearly outline the limitations of the proposed FL-GNN, and how these have been addressed, to provide a comprehensive perspective.

**Questions:**

* In addition to benchmarking performance, a detailed comparative analysis on how FL-GNN overcomes the limitations of standard GNNs would be insightful.

* The introduction of terms such as Type-2 fuzzy sets and intuitionistic fuzzy sets, which are mentioned as future work, might be unfamiliar to some readers. A brief description or reference would be helpful.

* Please provide more discussion on the related works [1,2]

[1] Fuzzy Representation Learning on Graph (https://ieeexplore.ieee.org/document/10061283)

[2] Graph Fuzzy System for the Whole Graph Prediction: Concepts, Models and Algorithms (https://ieeexplore.ieee.org/document/10287583)"

---

> ### Author Response · Authors · 2023-11-20
> **Thanks for the comments from Reviewer CtCE - part 1**
>
> We thank this reviewer for the constructive comments. We have carefully incorporated them in the revised paper. For clarity, we provide a detailed point-to-point response to each comment. Due to the space limit, we post our response in multiple comments labeled '- part 1', '- part 2', etc.
>
> Weakness 1 \& Question 1
> ======================
>
> **RE:** Thank you for your valuable suggestions. In the revised manuscript, we summarize the advantages of FL-GNN compared to traditional GNNs in Section 1 on page 2.
>
> **(1) Representation capacity.**
> Fuzzification has been extensively exploited to improve data representation capabilities. According to the Introduction Section and two supporting references [1,2], real-world data often contains uncertainty and fuzziness beyond the scope of traditional crisp values. To tackle this challenge, tools for fuzzification have been developed to capture such fuzziness and uncertainty. However, relying solely on fuzzification as a data augmentation technique is insufficient to fully utilize the benefits of fuzziness. Traditional GNN frameworks are limited in their feature extraction, which focuses solely on feature dimensions and disregards the fuzzy dimension. To overcome this limitation, we have created the FL-GNN, which utilizes fuzzy rules to highlight fuzzy-level features and leverages the advantages of fuzzy features for a more comprehensive data representation solution.
>
> **(2) Interpretability.**
> In traditional GNNs, the correlation between the network's parameter weights and inference process is implicit. In FL-GNN, the differences in topological structure between communities are explicitly presented in the firing strength distribution. By studying the firing strength distribution, we can know which rules are valid and which are redundant. Therefore, in this paper, interpretability not only provides a visual perspective for us to observe the local topology information and feature distribution differences of vertices but also provides reliable evidence for us to improve the model structure.
>
> In Appendix C, we provide additional experiments and analyses including: 1) Interpretability of indirect interactions between nodes. 2) Interpretability experiments conducted on the real-world dataset Pubmed. 3) The interpretability comparison between FL-GNN-A and GNNExplainer[3].
>
> **(3) Degree of freedom.**
> Several frameworks have been developed in graph inference, such as Message Passing Algorithms, which are compatible with most graph inference models. Fuzzy inference systems can offer a more flexible way to establish inference models. Fuzzy rules are not limited to graph-structured data and can be extended to other data structures and temporal data processing [4]. In addition, in [2], fuzzy rules are directly designed for specialized graph-level task inference. Therefore, we believe that the potential of FL-GNN will be fully realized in future works, where more effective fuzzy rules can be designed for graph inference tasks. In Appendix A, we have mentioned that the fuzzy rule of FL-GNN is limited to message aggregation in 1-order neighbors. Nevertheless, by designing more sophisticated fuzzy rules, they can be extended to complex topology structures such as "Hyperedge" and "Simplicial Complex", which will help the model break through the 1-dim WL-test.
>
> [1] [Fuzzy Representation Learning on Graph, IEEE Trans. Fuzzy Syst. 2023.](https://ieeexplore.ieee.org/document/10061283)
>
> [2] [Graph Fuzzy System for the Whole Graph Prediction: Concepts, Models and Algorithms, IEEE Trans. Fuzzy Syst. 2023.](https://ieeexplore.ieee.org/document/10287583)
>
> [3] [GNNExplainer: Generating Explanations for Graph Neural Networks, NeurIPS 2019.](https://arxiv.org/abs/1903.03894)
>
> [4] [An evolving recurrent interval type-2 intuitionistic fuzzy
> neural network for online learning and time series prediction, Appl. Soft Comput 2019.](https://doi.org/10.1016/j.asoc.2019.02.032)

---

> ### Author Response · Authors · 2023-11-20
> **Thanks for the comments from Reviewer CtCE - part2**
>
> Question 2 \& Weakness 2
> ======================
>
> **RE:** Thank you for your constructive suggestions. Regarding Question 2, we have added the differences between Type-2 fuzzy sets and intuitionistic fuzzy sets, as well as Type-1 fuzzy sets used in FL-GNN in Section 5 on page 10. Regarding weakness 2, we describe the aspects that may limit FL-GNN to reach its full potential in Section 5:
>
> 1\) FL-GNN uses Type-1 fuzzy sets to fuzzify the feature data of vertices, but the ability of Type-1 fuzzy sets to represent fuzziness is limited. Instead, more powerful fuzzy sets such as Type-2 fuzzy sets and intuitionistic fuzzy sets can provide stronger fuzziness. By using these fuzzy sets, we can obtain better representation capabilities.
>
> 2\) Computational complexity is relatively high given that FL-GNN needs to handle a large number of fuzzy rules. Though we proposed FL-GNN-A, which can greatly alleviate this problem and enable FL-GNN-A to work effectively on large datasets. One of our ongoing researches is to develop optimizations towards further improvement of the computational complexity. One direction we plan to pursue is to use the interpretability property of FL-GNN to guide us in designing evolutionary algorithms to compress network structures and obtain more effective models.
>
> Question 3
> ==========
>
> **RE:** We thank this reviewer for bringing the two relevant references [1, 2] to our attention. Both papers play a role in improving the quality of the related work.
>
> In [1], the authors propose using fuzzy features to carry out graph contrastive learning. The paper demonstrates the advantages of data fuzzification in representing graph information and supports this viewpoint with abundant experiments. However, the authors do not incorporate the core of fuzzy inference systems, i.e., the rule base and inference engine, into such graph inference tasks.
>
> In [2], the authors propose a model to deal with graph-level inference tasks, including graph regression tasks and graph classification tasks. The model refers to the fuzzy logic system to generate appropriate fuzzy rules to adapt specific inference tasks. The authors utilize the graph cluster algorithm to find the prototype graphs, and each prototype graph's feature will be used to generate an IF-part of the fuzzy rule. Then, the prototype graph will be fed into a GNN-dominated network structure GCPU to generate the network parameters for the THEN-part of the fuzzy rule. This work focuses on graph-level inference tasks but ignores the local topology information, while the model utilizes graph kernel function and traditional GNN to extract the graph information instead of fuzzy logic.
>
> In addition, we noticed that both papers acknowledged the importance of interpretability even though they failed to explore its advantages in graph inference tasks.
>
> In our work, we conducted a more in-depth study of the working principle and interpretability of FL-GNN. We revealed an interesting and explicit correlation between the local topological structure and firing strength distribution. Our model starts from node-level graph information, which allows FL-GNN to work in not only node-level inference tasks but also graph-level inference tasks. In Appendix C, we further provide evidence that the firing strength distribution can represent the whole graph information in the high-order layer by adopting a stacked FL-GNN structure. That is why FL-GNN could also work in graph-level inference tasks.
>
> We briefly introduce these two new references in the related work in Section 1 on page 2.
>
> [1] [Fuzzy Representation Learning on Graph, IEEE Trans. Fuzzy Syst. 2023.](https://ieeexplore.ieee.org/document/10061283)
>
> [2] [Graph Fuzzy System for the Whole Graph Prediction: Concepts, Models and Algorithms, IEEE Trans. Fuzzy Syst. 2023.](https://ieeexplore.ieee.org/document/10287583)
>
> Summary of the revision
> ==================
>
> - In Appendix C, we provide additional experiments and analyses including: 1) Interpretability of indirect interactions between nodes. 2) Interpretability experiments conducted on the real-world dataset Pubmed. 3) The interpretability comparison between FL-GNN-A and GNNExplainer.
> - In Section 1 on page 2, we highlight the limitations of traditional GNNs and explain how FL-GNN can address these limitations. Meanwhile, we replace the original presentation of related work with two mentioned papers that are more relevant to our work.
> - We have supplemented the experiments on the large-scale node-level dataset "ogbn-proteins" in Section 4.2.
> - In Section 5 on pages 9-10, we rewrite the conclusion by removing inadequate statements and discussing the current limitations of FL-GNN and possible ways to overcome them.

---

> > ### Comment · Reviewer_CtCE · 2023-11-23
> >
> > The authors have made a lot efforts to address the majority of my previous concerns, resulting in an improved version of the manuscript. However, I still have a few questions and suggestions that I believe could significantly enhance the paper's value and inspire future research endeavors:
> >
> > (1) I am intrigued by the possibility of extending the current framework to one based on Type-2 fuzzy sets. Could the authors elaborate on this potential extension and highlight the key challenges involved in adapting their framework to handle Type-2 fuzzy sets effectively?
> >
> > (2) While the paper has shown promising progress, the issue of stability remains a challenge, particularly when dealing with a large number of fuzzy rules. Could the authors delve further into strategies or techniques to mitigate the substantial time costs associated with handling a substantial number of fuzzy rules in practice?
> >
> >
> > I am inclined to raise my evaluation of the paper to 'marginally above the acceptance threshold' if the authors could provide more comprehensive responses and address the following comments in their revision.

---

> ### Author Response · Authors · 2023-11-23
> **Thanks for  the constructive comments by Reviewer CtCE - part 1**
>
> We are pleased to see that the majority of your concerns have been successfully addressed. We greatly appreciate your invaluable comments and suggestions, which have significantly improved the quality of our paper. Following your suggestions, we have carefully incorporated them in Section 5 of our revised paper. For clarity, we provide a detailed point-to-point response to each question. Due to the space limit, we post our response in multiple comments labeled '- part 1', '- part 2', etc.
>
>
> Question 1
> ==
>
> **RE:** FL-GNN is compatible with Type-2 fuzzy sets. Here, we use a simple example to illustrate. Usually in FNN, in order to reduce the computational complexity, we use interval Type-2 fuzzy sets [1]. In interval Type-2 fuzzy sets, we can use two Type-1 fuzzy sets, i.e., lower membership function (LMF) and upper membership function (UMF), to characterize the upper and lower bounds of the membership function, respectively. The second-order fuzziness of interval Type-2 fuzzy sets is reflected by the gap between UMF and LMF, i.e., the footprint of uncertainty (FOU). If the FOU disappears, the interval Type-2 fuzzy sets will degenerate into Type-1 fuzzy sets. For the FL-GNN that introduces interval Type-2 fuzzy sets, we only need to calculate the firing strength of each fuzzy rule on LMF and UMF to obtain $\overline{\pmb{r}_n}=[\overline{r^1_n}, \overline{r^2_n}, \dots, \overline{r^K_n}]$ and $\underline{\pmb{r}_n}=[\underline{r^1_n}, \underline{r^2_n}, \dots, \underline{r^K_n}]$ at the rule layer, where $n$ is the index of the node in the graph and $K$ is the number of fuzzy rules. $\overline{\pmb{r}_n}$ is the firing strength vector for UMF and  $\overline{r^k_n}$ is the firing strength of the $k$th fuzzy rule of UMF, $\underline{\pmb{r}_n}$ is the firing strength vector for LMF and $\underline{r^k_n}$ is the firing strength of the $k$th fuzzy rule of LMF.
> Next, we can simply calculate the median value $\pmb{r}_n= \frac{\overline{\pmb{r}_n} + \underline{\pmb{r}_n}}{2}$ as the final firing strength of the $k$th fuzzy rule. Alternatively, we can directly concatenate $\pmb{r}_n= [\overline{\pmb{r}_n}, \underline{\pmb{r}_n}]$ and input them into the MLP to transform them into the form required by the downstream task, because $\overline{\pmb{r}_n}$ and $\underline{\pmb{r}_n}$ both possess the node feature information and graph topology information needed for graph inference.
>
> With the introduction of the Type-2 fuzzy set, the complexity of the model will be further increased compared to the Type-1 fuzzy set. The key challenge is how to reduce the computational complexity of the model. Meanwhile, to better capture the second-order fuzziness of the data, how to set the initial FOU of UMF and LMF and maintain the effective FOU during model training are also the key challenges we will face.
>
> [1] [Type-2 Fuzzy Logic: Theory and Applications, 2008](https://dblp.org/rec/books/sp/CastilloM08.html?view=bibtex)

---

> ### Author Response · Authors · 2023-11-23
> **Thanks for the constructive comments by Reviewer CtCE - part 1**
>
> Question 2
> ==
>
> **RE:** We can reduce the number of fuzzy rules of the model in the following three ways:
>
> **1\) Data dimensionality reduction.** Reducing data dimensionality is the most effective way to decrease the number of fuzzy rules. In the field of GNN, some excellent graph embedding methods [1,2] can compress data dimensionality while preserving graph information. Similarly, in the field of FNN, some online methods of data dimensionality reduction, such as the incremental deep pretraining (IDPT) algorithm [3], can be used to dynamically adjust hidden layer dimensions during model training based on a Restricted Boltzmann Machine.
>
> **2\) Highly efficient fuzzy rules.** This paper introduces a node-level message fuzzification approach for general graph inference tasks. In practice, more efficient fuzzification rules can be designed for specific tasks. For example, in [4], the authors directly implemented graph-level message fuzzification, which is more efficient for graph-level reasoning tasks. In [5], the OR-AND fuzzy rule structure was designed to directly avoid generating a large number of fuzzy rules.
>
> **3\) Evolutionary network structure.** Evolutionary network structures have been extensively studied in the field of FNN. A method based on improved density clustering was proposed in [6] to control the generation of fuzzy rules. Another study [7] introduced a self-organization network structure where the growth and branch reduction of fuzzy rules are controlled by identifying covariate shifts. In the future, we aim to improve the generation of fuzzy rules by identifying the effectiveness of fuzzy rules for inference, thus effectively reducing the generation of redundant fuzzy rules.
>
> In addition, some useful tricks can also be employed to enhance the efficiency of the model. For example, by using message pre-propagation, we can store the aggregated high-level node interaction information beforehand and then perform fuzzification in the GPU, which can significantly reduce the memory overhead.
>
> [1] [Complex Embeddings for Simple Link Prediction, ICML 2016.](https://proceedings.mlr.press/v48/trouillon16.pdf)
>
> [2] [GraphAIR: Graph Representation Learning with Neighborhood Aggregation and Interaction, Pattern Recognit. 2021](https://arxiv.org/abs/1911.01731)
>
> [3] [An Efficient Self-Organizing Deep Fuzzy Neural Network for Nonlinear System Modeling, IEEE Trans. Fuzzy Syst. 2022.](https://ieeexplore.ieee.org/document/9423589)
>
> [4] [Graph Fuzzy System for the Whole Graph Prediction: Concepts, Models and Algorithms, IEEE Trans. Fuzzy Syst. 2023.](https://ieeexplore.ieee.org/document/10287583)
>
> [5] [Disjunctive Fuzzy Neural Networks: A New Splitting-Based Approach to Designing a T–S Fuzzy Model, IEEE Trans. Fuzzy Syst. 2022.](https://ieeexplore.ieee.org/document/9264752)
>
> [6] [An evolving recurrent interval type-2 intuitionistic fuzzy neural network for online learning and time series prediction, Appl. Soft Comput. 2019.](https://www.sciencedirect.com/science/article/pii/S1568494619300961)
>
> [7] [Self-Organizing Interval Type-2 Fuzzy Neural Network With Adaptive Discriminative Strategy, IEEE Trans. Fuzzy Syst. 2023.](https://ieeexplore.ieee.org/abstract/document/9930109)

---

### Official Review · Reviewer_bUeC · 2023-11-01

**Soundness:** 3 good
**Presentation:** 3 good
**Contribution:** 2 fair
**Rating:** 6
**Confidence:** 3

**Summary:**

This paper proposes a hybrid fuzzy-logic graph neural network by combining fuzzy neural network with GNN to effectively capture and aggregate local information flows within graph structural data. FL-GNN by design has three novel features. The structure fuzzy rule is to boost the graph inference capability. The interpretability is enhanced by adding the analytic exploration methods to its graph inference. The experiments show the improvements over several baselines.

**Strengths:**

1. This paper focuses on an interesting problem to the graph machine learning community, which is to combine fuzzy neural network with GNN.
2. The paper is well written. The framework is clear to present the details of the proposed method, especially Figure 1 and 2.
3. The experiments are conducted on real-world datasets such as OGB.

**Weaknesses:**

1. Some claims need to be further validated. The authors argue that the interpretability of FL-GNN is good while they ignore the experiments to compare with some graph interpretability baselines (please refer to [1]).
2. The experiments in terms of node-level predictions should consider more large-scale datasets from OGB, just as the authors have done in graph-level predictions.
3. Typos: dataset name Coar -> Cora in section 4.1.

**Questions:**

See weaknesses

---

> ### Author Response · Authors · 2023-11-20
> **Thanks for the comments from Reviewer bUeC**
>
> We thank this reviewer for the constructive comments, which are very helpful for us to improve our paper. We have carefully incorporated them in the revised paper. For clarity, we provide a detailed point-to-point response to each comment, followed by a summary of the revision.
>
> Weakness 1
> ==========
>
> **RE:**  Thanks for your comments and suggestions. FL-GNN's interpretability is inherent in fuzzy inference systems. The paper shows formally and empirically that FL-GNN can retain this interpretability in graph inference in specific scenarios.
>
> Following your suggestion, we provide additional experiments and analysis in Appendix C to validate our claims on interpretability, including: 1) Interpretability of indirect interactions between nodes; 2) Interpretability experiments conducted on the real-world dataset Pubmed; 3) Interpretability comparison between FL-GNN-A and GNNExplainer [1].
>
> [1] [GNNExplainer: Generating Explanations for Graph Neural Networks, NeurIPS 2019.](https://arxiv.org/abs/1903.03894)
>
> Weakness 2
> ==========
>
> **RE:** Thanks for your suggestion. We have provided an experiment on a large-scale node-level dataset "ogbn-proteins" in Section 4.2 of the revised manuscript. The new experimental results are displayed in Table 2-1 (a part of Table 1 in the revised manuscript), showing that our model achieved comparable results to traditional GNNs.
>
> &emsp;&emsp;&emsp;&emsp;&emsp;&emsp;&emsp;&emsp;&emsp;&emsp;&emsp;&emsp;&emsp;&emsp;&emsp;&emsp;**Table 2-1: The experimental results on ogbn-proteins**
>
> |FL-GNN-A|Graphsage-mean|GAT|GCN|TransformerConv|GATv2|GIN|
> |------------|------------|------------|------------|------------|------------|------------|
> |0.7989&pm;0.0043| 0.7962&pm;0.0039|0.7996&pm;0.0018|0.7551&pm;0.0079 |0.8003&pm;0.0025|0.7908&pm;0.0033|0.7569&pm;0.0054|
>
> Weakness 3
> ==========
>
> **RE:**
> Thank you for your careful review and attention to detail. We sincerely apologize for any inconvenience caused by these typos. We thoroughly revised the article to address these typos and ensure its accuracy and clarity. We proofread the manuscript to correct these typos, e.g.,
>
> - On page 1: "fields such as medical image process" $\rightarrow$  FNN has been successfully applied in various fields such as medical image processing.
> - On page 2: "information are ubiquitous" $\rightarrow$ This sentence was deleted after reorganizing this part.
> - In experiments: "Coar" $\rightarrow$ "Cora".
>
> Summary of the revision
> ================
>
> - In Section 1 Introduction, we conclude the limitations of current GNNs and indicate how FL-GNN compensates for these shortcomings. Meanwhile, we replace the original presentation of related work with two relevant papers [1,2].
> - We have supplemented an experiment on a more large-scale dataset in Section 4.2.
> - We proofread our manuscript and corrected incorrect formulas and typos.
>
> [1] [Fuzzy Representation Learning on Graph, IEEE Trans. Fuzzy Syst. 2023.](https://ieeexplore.ieee.org/document/10061283)
>
> [2] [Graph Fuzzy System for the Whole Graph Prediction: Concepts, Models and Algorithms, IEEE Trans. Fuzzy Syst. 2023.](https://ieeexplore.ieee.org/document/10287583)

---

### Official Review · Reviewer_Wx81 · 2023-11-10

**Soundness:** 2 fair
**Presentation:** 2 fair
**Contribution:** 2 fair
**Rating:** 6
**Confidence:** 3

**Summary:**

The authors propose a combination of Takagi-Sugeno Fuzzy Neural Networks (TS-FNNs) and Graph Neural Networks (GNNs), which they call a Fuzzy-Logic Graph Neural Network (FL-GNN).
The FL-GNN architecture consists of five layers: A fuzzification layer and a rule layer, followed by normalization, defuzzification and output layers.
First, the fuzzification layer assigns multiple fuzzy set membership degrees to each vertex feature of a given input graph.
Then, in the rule layer, the features of neighboring vertices are combined via learned fuzzy-logic rules whose firing strength depends on the degree to which features of adjacent vertices are members of the same fuzzy sets; the structure of those fuzzy rules is inspired by the message passing algorithm (MPA), that is typically used in GNNs.
The following three layers then transform the results of the fuzzy rules into a final prediction for each vertex.

Since the proposed architecture for FL-GNNs turns out to be computationally expensive, the authors also propose a simplified variant, called FL-GNN-A, which uses sliding windows and max pooling to significantly reduce the number of rules.

In the conducted experiments, the FL-GNN-A model was compared against multiple other state-of-the-art GNN architecture on a selection of seven common node-level and graph-level benchmark datasets.
Overall FL-GNN-A appears to perform similarly to the models it was compared against.
In the conclusion the authors state that the proposed combination of fuzzy inference and GNNs can improve interpretability and offer new insights into graph inference.

**Strengths:**

First, the paper is well structured and provides a good introduction to fuzzy neural networks, without assuming an in-depth knowledge about fuzzy logic and fuzzy inference.
The language is clear and the provided figures support the written explanations well and are easy to understand.

Second, the idea of combining fuzzy logic with message passing and the potential to improve model interpretability via fuzzy rules is interesting.
The example in appendix C, showing that meaningful rules can be learned via FL-GNNs, is promising.

**Weaknesses:**

I see the following four weaknesses, ordered descendingly by importance:

First, the paper does not provide convincing evidence that the proposed FL-GNN architecture has consistent, measurable advantages over existing GNN architectures.
The experimental results show that FL-GNNs are on-par with previously proposed approaches.
While the authors allude to potential insights into graph inference and improved model interpretability in the conclusion, the anecdotal evidence provided in appendix C is, by itself, not sufficient to claim that FL-GNNs provide interpretable predictions.
Here, a more comprehensive evaluation and a comparison with general gradient-based approaches, such as [GradCAM](https://arxiv.org/abs/1610.02391), or graph-specific approaches, such as [GnnExplainer](https://arxiv.org/abs/1903.03894) would have been interesting.
As it stands, the relevance of the proposed architecture is unclear to me.

Second, the definition of the FL-GNN architecture in section 3 is, in parts, lacking formal accuracy. Here are a few examples:
- Section 3.1: The signatures of $\phi$ and $\psi$, as well as those of $\rho$ and $\sigma$ are unclear. The authors state that the signature of $\phi$ is $V \to F_v$ but also that $\phi(v_i) \subset F_v$. Similarly, supposedly $\rho(\phi(v_i)) \subset A_v$, even though the target domain of $\rho$ is $A_v$, not $2^{A_v}$.
- Section 3.2, eq. 10: Why is the "AND" in the condition taken over all $A_m \in \{ A_{m_{1,i}},\dots, A_{m_{D,j}} \}$? Since $i$ is used to denote the feature index and $j$ to denote the index of a fuzzy  subset, $A_{m_{1,j}}$ was probably meant here. Is this correct?
- Section 3.2: $S$ is defined as the Cartesian product of $D$ sets of size $M$, therefore $|S| = M^D$; in the last line on page 4 the authors do however state that $S$ has $M^D$ subsets.
Even though each of those examples can be considered to be minor formal mistakes, the repeated unclear usage of "element" and "subset" of a set make it difficult to understand the precise meaning of the provided formal definitions.

Third, the proposed approach only considers the direct neighbors of a vertex in each rule. Indirect interactions between vertices therefore are not captured by FL-GNNs, as also alluded to by the authors at the end of appendix C.
The authors do not discuss if and how meaningful interpretations could be provided in problem domains where such indirect interactions are important.

Last, I noticed a number of minor typos, that could have easily been prevented by proofreading. A few examples: Consistently used "Coar" instead of "Cora" in section 4, "fields such as medical image process" (page 1), "information are ubiquitous" (page 2).

**Questions:**

1. Is there additional evidence supporting the claim that FL-GNNs can provide meaningful interpretations? Is the quality of those interpretations/explanations better than that of other XAI approaches?
2. Concerning the formal definitions, can you clarify the intended meaning of the mentioned formulas?
3. Can FL-GNNs be adapted to also capture indirect vertex interactions and, if so, how?

---

> ### Author Response · Authors · 2023-11-20
> **Thanks for the comments from Reviewer Wx81- part 1**
>
> We thank this reviewer for the constructive comments. We have carefully incorporated them in the revised paper. For clarity, we provide a detailed point-to-point response to each comment. Due to the space limit, we post our response in multiple comments labeled '- part 1', '- part 2', etc.
>
> Weakness 1 \& Question 1
> =
> **RE:** One of the main contributions made by FL-GNN is its interpretability inherent in fuzzy inference systems. This paper mainly intends to demonstrate that FL-GNN can retain this interpretability in graph inference in specific scenarios from the following three aspects: the meaning and the potential value of FL-GNN interpretability, and the comparative discussion with the existing representative methods, such as [1-2] provided by the reviewer.
>
> **(1) The meaning of FL-GNN's interpretability.**
>
> First, as discussed in Appendix C, in a social network scenario where the raw features of each vertex are comprehensible, with FL-GNN, one can confer the feature with meaningful linguistic labels to make the fuzzy rule intuitively understood.
>
> Second, the interpretability of FL-GNN is not limited to linguistic labels and can capture certain meanings based on hidden features. That is, the firing strength distribution of the fuzzy rule can provide interpretations for the inference process of FL-GNN. The firing strength distribution of communities directly reflects their topological structure information and feature distribution. Here, topological information is specified as the degree of community overlap, i.e., the probability of having edges within a community and between multiple communities. Feature distribution specifically refers to the differences in node feature value between different communities. As shown in Appendix C, the communities of different professions exhibit intuitive differences in firing strength distribution.
>
> **In FL-GNN, the differences in topology structure between communities are explicitly presented in the firing strength distribution, indicating that there is a direct relationship between which fuzzy rules have a positive effect on capturing topology differences and which ones do not work.** Therefore, our improvement in fuzzy rules can be intuitively reflected in the differences in model performance, just as we performed MaxPooling on the output of fuzzy rule layers in FL-GNN-A. Filtering out redundant rules not only ensures model accuracy but also improves the inference speed of the model.
>
> **(2) The potential value of FL-GNN's interpretability.**
>
> We believe that studying firing strength distributions has significant potential values because the fuzzy rule provides a medium that allows us to indirectly explain the working principles of neural networks by directly observing their intermediate results. The firing strength distributions can make uncovering flaws in the network structure or the dataset easier. Here, we provide several possible explorations that can take advantage of the FL-GNN's interpretability.
>
> 1\) Outlier detection: For example, if there exist some nodes that have a significant difference in their firing strength distributions compared to their communities, or two different communities present resemble firing strength distributions, then there might be misclassification or overclassification of labels in this dataset, and the dataset may also have higher-order topological differences that the current fuzzy rule cannot capture.
>
> 2\) Optimizing network structure: In this paper, we have also made a simple attempt to use firing strength distribution to optimize the structure of FL-GNN. In part-2 of FL-GNN-A, we add the MaxPooling layer after observing redundant low firing strength fuzzy rules. In addition, we are considering combining interpretability with evolutionary algorithms to adjust network architecture and make it more efficient.

---

> ### Author Response · Authors · 2023-11-20
> **Thanks for the comments from Reviewer Wx81 - part 2**
>
> **(3) The interpretability comparison between FL-GNN and other related works.**
>
> Thanks for bringing to our attention the two relevant papers [1,2]. We have provided the interpretability comparison experiments between FL-GNN and them in Appendix C.
>
> GradCAM is a technique that uses network gradients to locate important areas that affect the output decisions of CNN models. In comparison, FL-GNN primarily focuses on enhancing the interpretability of GNN models. Due to the time constraint of Rebuttal, we have not come up with a suitable interpretability measurement method for comparison with FL-GNN, so we may not have time to include GradCAM in the comparison experiments.
>
> GnnExplainer is a model-free method that can provide a reliable explanation for GNN that conforms to the Message Passing paradigm. Specifically, GnnExplainer will take a trained GNN model and its prediction results as input, and output the subgraphs and feature sets that truly affect the prediction results.
>
>
> In GnnExplainer, the main factors affecting model predictions are obtained through backward analysis of the predicted results. Different from GnnExplainer, the interpretability of FL-GNN is present in its intermediate calculation results during forward inference, namely the firing strength distribution. The firing strength distribution is clearly related to the model's capability to capture topological structure information and feature distribution. Similar to GnnExpander, FL-GNN can also evaluate which fuzzy rules have a positive effect on inference through firing strength distribution and which rules are redundant. Though FL-GNN cannot directly explain which subgraph plays a crucial role in model inference, we can still indirectly discover useful topology and feature information such as community overlap degree and feature distribution similarity by observing the firing strength distribution between communities. In contrast, although GnnExplainer can provide model-free explanatory factors, the intermediate behavior in the model inference process remains opaque like a black box. FL-GNN provides explanatory behavior based on the reasoning process of the model, which is more intuitive for users to understand the working principle of the model.
>
> In Appendix C, we conducted a fidelity testing experiment to confirm that fuzzy rules with low firing strength values are redundant for the FL-GNN inference process. This also demonstrates that the firing strength distribution is directly linked to the graph inference process of FL-GNN. Specifically, we conducted two groups of experiments. In the first group, we pre-trained the FL-GNN-A on the Pumbed dataset and marked the indexes of the fuzzy rules with low firing strengths during the inference phase of the pre-trained model. Then, we re-trained the model while masking out the corresponding fuzzy rules for the bottom 70\%, 50\%, and 30\% of the ranked firing strength values in each layer. In the second group, we fed the FL-GNN-A model pre-trained on the Pubmed dataset into the GNNExplainer model to extract the subgraphs and feature sets more useful for the inference task. We then used the information extracted by GNNExplainer for inference on the test dataset to calculate the fidelity of GNNExplainer. The regularization hyperparameters for subgraph size were 0.005, for laplacian were 0.5, and for feature explanation were 0.1.
>
> The experimental results are presented in Table 1-1 (Table 6 in the revised manuscript). We found that masking rules with low firing strength only slightly affect the model's performance, indicating that rules with low firing strength are redundant to the inference process. Additionally, we found that the information extracted by GNNExplainer significantly improves the inference performance. This is because the GNNExplainer is trained based on node labels, but masking redundant rules does not introduce additional label information.
>
> &emsp;&emsp;&emsp;&emsp;&emsp;&emsp;**Table 1-1: The Fidelity Experiment on the Pubmed dataset.**
> | Pumbed | Pubmed-70\% | Pubmed-50\% | pubmed-30\% | GNNExplainer |
> |------------ |-------------------- |---------------------|--------------------|---------------------|
> |0.7704 &pm; 0.0012|0.7536 &pm; 0.0090|0.7733 &pm; 0.0013|0.7803 &pm; 0.0019|0.9397 &pm; 0.0004|
>
>
> [1] [GNNExplainer: Generating Explanations for Graph Neural Networks, NeurIPS 2019.](https://arxiv.org/abs/1903.03894)
>
> [2] [Grad-CAM: Visual Explanations from Deep Networks via Gradient-based Localization, ICCV 2017.](https://arxiv.org/abs/1610.02391)

---

> ### Author Response · Authors · 2023-11-20
> **Thanks for the comments from Reviewer Wx81 - part 3**
>
> Weakness 2 \& Question 2
> =
>
> **RE:** Thanks for your helpful comments. We have corrected the errors in the revised manuscript.
>
> Subquestion 1
> -
>
> **RE:** The mapping functions were defined inaccurately, and we corrected them as follows:
>
> Let $\phi: V \rightarrow 2^{F_v}$, $\psi: E \rightarrow 2^{F_e}$ denote the mapping functions for vertex attributes and edge attributes, where $2^{F_v}$ and $2^{F_e}$ are the power sets of $F_v$ and $F_e$, respectively,
> and then each vertex $v_i \in V$ and edge $e_i \in E$ can be represented by a set of vertex attributes $\phi(v_i) \subset F_v$ and a set of edge attributes $\psi(e_i)\subset F_e$. Meanwhile, we also define two fuzzy subset mapping functions, $\rho: 2^{F_v} \rightarrow 2^{A_v}, \sigma: 2^{F_e} \rightarrow 2^{A_e}$.
>
> Subquestion 2
> -
>
> **RE:** For the first question, the calculation result of two fuzzy subsets by using logic conjunction operation ("AND") or logic disjunction operation ("OR") is still a fuzzy subset, and the expression $v\ is\ A_m$ is a procession of fuzzification that also generates a fuzzy subset.
> the expression $v \ is \ A_m$ represents a fuzzy subset generated by fuzzification. In the IF-part of Eq. 10, all operations are performed between fuzzy subsets, so it is reasonable to use the "AND" operation to combine a set of fuzzy subsets. This mathematical expression is commonly used in papers on fuzzy logic [1,2,3] (some papers may use $\bigwedge$ or $T(\cdot)$ to denote the "AND" operation).
>
> For the second one,  we apologize for misusing indexes $i$ and $j$ in Eq. 10. The indexes $i$ and $j$ in Eq. 10 are unrelated to the feature index and the fuzzy subset index, and they denote one index combined result generated from the Cartesian product. To avoid confusion, we substitute $i$ and $j$ in Eq. 10 with other irrelevant symbols (e.g., $a$ and $b$) in the revised manuscript.
>
> [1] [Disjunctive Fuzzy Neural Networks: A New Splitting-Based Approach to Designing a T–S Fuzzy Model, IEEE Trans. Fuzzy Syst. 2020.](https://ieeexplore.ieee.org/document/9264752/)
>
> [2] [Hybrid Learning for Interval Type-2 Intuitionistic Fuzzy Logic Systems as Applied to Identification and Prediction Problems, IEEE Trans. Fuzzy Syst. 2018.](https://ieeexplore.ieee.org/document/8286852)
>
> [3] [Deep Takagi–Sugeno–Kang Fuzzy Classifier With Shared Linguistic Fuzzy Rules, IEEE Trans. Fuzzy Syst. 2017.](https://ieeexplore.ieee.org/document/7984865)
>
>
> Subquestion 3
> -
> **RE:** We apologize for the minor formal mistakes, where the result of n-ary Cartesian product should be denoted as a set of n-tuples, not subsets. Here, we present an example to clarify this problem. Assuming that we have three sets of fuzzy subsets and each set has three fuzzy subsets $A_{v,1}=\\{A_{m_{1,1}},A_{m_{1,2}},A_{m_{1,3}} \\},A_{v,2}=\\{A_{m_{2,1}}, A_{m_{2,2}}, A_{m_{2,3}}\\}, A_{v,3}=\\{A_{m_{3,1}},A_{m_{3,2}}, A_{m_{3,3}}\\}$, to use the 3-ary Cartesian product to combine these three sets, we obtain: $S=A_{v,1}\times A_{v,2}\times A_{v,3}=$$\\{$$(A_{m_{1,1}},A_{m_{2,1}},$$A_{m_{3,1}})$$,(A_{m_{1,2}},A_{m_{2,1}},A_{m_{3,1}})$,...,$(A_{m_{1,3}}$$,A_{m_{2,3}}$,$A_{m_{3,3}}$)}, $|S|=8$. Each element in $S$ is a tuple such as $(A_{m_{1,1} },A_{m_{2,1}},A_{m_{3,1}})$. We have rewritten inappropriate expressions in the revised manuscript.
>
> Weakness 3 \& Question 3
> =
>
> **RE:** Thank you for bringing up the issue of node indirect interaction. We would like to address your concern and clarify that FL-GNN is designed to effectively capture the indirect interaction of nodes (see Appendix A for details). To achieve higher-order reasoning ability, FL-GNN utilizes a stacked structure. Additional experiments have been included in Appendix C to better showcase how FL-GNN can capture indirect interactions between nodes. We have also demonstrated the meaningful interpretations of the firing strength distribution generated by nodes under high-order interactions. Specifically, we have visualized the firing strength distribution of each layer in FL-GNN with the stacked structure. By observing the changes in trends of firing strength distribution in different communities, we have found that as the interaction order increases, the firing strength distribution of a community gradually changes from reflecting the topology information of a single community to that of multiple overlapping communities. Ultimately, the firing strength distributions of all communities will tend to be consistent, representing the topology information of the entire graph.
>
> Weakness 4
> =
>
> **RE:** We appreciate your diligent examination. We are sorry for any inconvenience caused by these typos. We proofread the manuscript and correct the typos, e.g.,
>
> - On page 1: "fields such as medical image process" &rarr;  FNN has been successfully applied in various fields such as medical image processing.
>
> - On page 2: "information are ubiquitous" &rarr; This sentence was deleted after reorganizing this part.
>
> - In experiments: "Coar" &rarr; "Cora".

---

> ### Author Response · Authors · 2023-11-20
> **Thanks for the comments from Reviewer Wx81 - part 4**
>
> **Summary of the revision**
>
> - In Appendix C, we provide additional experiments and analyses to include: 1) Interpretability of indirect interactions between nodes. 2) Interpretability experiments conducted on the real-world dataset Pubmed. 3) The interpretability comparison between FL-GNN-A with GNNExplainer.
>
> - We proofread our manuscript and corrected incorrect formulas and typos.
>
>
> [1] [Fuzzy Representation Learning on Graph, IEEE Trans. Fuzzy Syst. 2023](https://ieeexplore.ieee.org/document/10061283)
>
> [2] [Graph Fuzzy System for the Whole Graph Prediction: Concepts, Models and Algorithms, IEEE Trans. Fuzzy Syst. 2023](https://ieeexplore.ieee.org/document/10287583)

---

> > ### Comment · Reviewer_Wx81 · 2023-11-20
> >
> > Thank you for the detailed response and the added experimental results!
> > My questions are now mostly answered; however, one aspect I did not fully understand based on the description provided in the paper is the stacking of layers in FL-GNNs.
> >
> > What exactly is meant by the term "layer" in Figure 6? I assume that multiple rule layers were stacked on top of each other but I did not find a formal explanation of how this stacking works.

---

> > > ### Author Response · Authors · 2023-11-21
> > > **Thanks for the prompt reply from Reviewer Wx81**
> > >
> > > Thanks very much for your quick response and constructive comments. We have added relevant definitions in Appendix C (on page 17).
> > > The meaning of "Layer" in Figure 6 is the number of stacked layers of FL-GNN, and the stacked structure is a stack of multiple FL-GNNs rather than a stack of rule layers, where each layer corresponds to an FL-GNN. Similarly, the stacked structure of FL-GNN-A is stacking the entire FL-GNN-A.
> > >
> > > In Section 3.2, we take Multiple Input Single Output (MISO) FL-GNN as an example to illustrate. Here, we use Multiple Input Multiple Output (MIMO) FL-GNN for the generic case to introduce stacked FL-GNNs. The output of MIMO FL-GNN is defined as $\pmb{O}^5\in\mathbb{R}^{N\times hidden\underline{}nums}$, where $N$ denotes the number of nodes. The difference between MISO FL-GNN and MIMO FL-GNN is that the value space of trainable parameters $\pmb{q}^{k}$ of $k$th fuzzy rule changes from $\mathbb{R}^{K}$ to $\mathbb{R}^{K \times hidden\underline{}nums}$, where $hidden\underline{}nums$ is the hidden dimension size.
> > >
> > > We formally define the FL-GNN with a stacked structure as follows: First, similar to the definition of FL-GNN in Section 3.2, a single-layer MIMO FL-GNN is defined as $\pmb{O}^5 = f_\theta(\pmb{X}, \pmb{A})$, where $\pmb{X}$ denotes the input node data, $\pmb{A}$ denotes the adjacency matrix, $\theta$ denotes the trainable parameters in FL-GNN, and $\pmb{O}^5$ denotes the output of the single layer of FL-GNN. In the following description, we replace $\pmb{O}^5$ with $\pmb{H}$ to make the formulas more concise.
> > > Then, for the FL-GNN with a stacked structure, we define the output of the $l$th layer as $\pmb{H}^{l}= f_{\theta^{l}}(\pmb{H}^{l-1}, \pmb{A})+\pmb{H}^{l-1}$, where we add the residual connection $\pmb{H}^{l-1}$. Meanwhile, some research papers have reported that the FNN structure has the ability to fit nonlinear functions [1, 2, 3]. To enhance the sparsity of the parameters and mitigate overfitting, we incorporate an activation function $\sigma(\cdot)$ of the ReLU family in each layer of FL-GNN. In addition, we also add a BatchNormalization Layer $\text{BN}(\cdot)$ for the stability of the model. Finally, the output of the $l$th layer is defined as $\pmb{H}^{l}= \text{BN}(\sigma(f_{\theta^{l}}(\pmb{H}^{l-1}, \pmb{A})+\pmb{H}^{l-1}))$.
> > >
> > > [1] [Fuzzy Representation Learning on Graph, IEEE Trans. Fuzzy Syst. 2023](https://ieeexplore.ieee.org/document/10061283)
> > >
> > > [2] [FRULER: Fuzzy Rule Learning through Evolution for Regression, Inf. Sci. 2016](https://www.sciencedirect.com/science/article/pii/S0020025516301591?via%3Dihub)
> > >
> > > [3] [ANFIS: adaptive-network-based fuzzy inference system, IEEE Trans. Syst. Man Cybern 1993](https://ieeexplore.ieee.org/stamp/stamp.jsp?tp=&arnumber=256541)

---

> > > > ### Comment · Reviewer_Wx81 · 2023-11-22
> > > >
> > > > Thank you for the explanation and the additional revision; this makes things a lot clearer.
> > > >
> > > > Overall, I am still not fully convinced that the proposed method provides meaningful advantages over existing GNN architectures, considering both its predictive quality and the quality of explanations one might derive from the rules.
> > > > Nonetheless, the proposed idea of combining FNNs and GNNs is interesting and might inspire further research in that direction.
> > > > My concerns about the formal definition of FL-GNNs were resolved in the revised version.
> > > >
> > > > Therefore, I change my rating to "marginally above the acceptance threshold".

---

> > > > > ### Author Response · Authors · 2023-11-23
> > > > > **Thanks for the constructive comments by Reviewer Wx81**
> > > > >
> > > > > We are pleased to see that your concerns have been successfully addressed. We would like to express our gratitude for your dedicated time and effort in scrutinizing our paper and providing invaluable feedback.
> > > > >
> > > > > As you pointed out, our main motivation/contribution is to generalize FNNs to graph inference tasks by combining the structure of FNNs with the design concepts of GNNs, which will inspire new research in this area. Currently, FNNs have been applied to various areas, but there is less research on processing graph data. FNN structures have a high degree of freedom, especially in defining fuzzy rules, making them versatile for a wide range of inference tasks. This property allows us to develop models that are more specific to the problem at hand. Moreover, FNNs are naturally interpretable and can handle fuzzy-level information, which is often not available in traditional deep-learning models. These properties not only help us understand the model behavior but also enhance FNNs' representational capabilities.

---

### Meta-Review · Area_Chair_pJRC · 2023-12-09

**Metareview:**

This paper develops a hybrid fuzzy-logic graph neural network method that combines fuzzy neural networks with GNN. The authors' goal is to effectively capture and aggregate local information flows within graph structural data. The authors emphasize several novel features of FL-GNN, including improved graph inference capability based on fuzzy rules and enhanced interpretability. Promising experimental results are also presented.

Although the reviewers raise a number of critical points in their original reports, there is agreement that the paper has potential. The authors showed a high level of commitment during the rebuttal phase and did their best to respond to the comments and to improve the submission. This was appreciated and positively acknowledged by all. In the discussion between authors and reviewers, some critical points could be resolved and some questions clarified. Other points remained open and were critically reconsidered in the subsequent internal discussion. The paper is still considered somewhat borderline, and the reviewers are still not fully convinced that the proposed method provides meaningful advantages over existing GNN architectures in terms of predictive quality and  interpretability. Eventually, the submission was found to be remain a bit behind the expectations for a top venue such as ICLR.

**Justification For Why Not Higher Score:**

Advantages over standard GNNs not convincingly demonstrated.

**Justification For Why Not Lower Score:**

N/A

---

### Decision · Program_Chairs · 2024-01-16

Reject